# ON THE BENEFITS OF MAXIMUM LIKELIHOOD ESTIMATION FOR REGRESSION AND FORECASTING

**Pranjal Awasthi, Abhimanyu Das, Rajat Sen & Ananda Theertha Suresh**
Google Research
{pranjalawasthi, abhidas, senrajat, theertha}@google.com

## ABSTRACT

We advocate for a practical Maximum Likelihood Estimation (MLE) approach towards designing loss functions for regression and forecasting, as an alternative to the typical approach of direct empirical risk minimization on a specific target metric. The MLE approach is better suited to capture inductive biases such as prior domain knowledge in datasets, and can output post-hoc estimators at inference time that can optimize different types of target metrics. We present theoretical results to demonstrate that our approach is competitive with any estimator for the target metric under some general conditions. In two example practical settings, Poisson and Pareto regression, we show that our competitive results can be used to prove that the MLE approach has better excess risk bounds than directly minimizing the target metric. We also demonstrate empirically that our method instantiated with a well-designed general purpose mixture likelihood family can obtain superior performance for a variety of tasks across time-series forecasting and regression datasets with different data distributions.

## 1 INTRODUCTION

The task of fitting a regression model for a response variable $y$ against a covariate vector $\boldsymbol{x} \in \mathbb{R}^d$ is ubiquitous in supervised learning in both linear and non-linear settings (Lathuilière et al., 2020; Mohri et al., 2018) as well as non-i.i.d settings like multi-variate forecasting (Salinas et al., 2020; Wang et al., 2019). The end goal in regression and forecasting problems is often to use the resulting model to obtain good performance in terms of some *target metric* of interest on the population level (usually measured on a previously unseen test set). The mean-squared error or the mean absolute deviation are examples of common target metrics.

In this paper, our focus is on the choice of loss function that is used to train such models, which is an important question that is often overlooked, especially in the deep neural networks context where the emphasis is more on the choice of network architecture (Lathuilière et al., 2020).

Perhaps the most common method used by practitioners for choosing the loss function for a particular regression model is to directly use the *target metric* of interest as the loss function for *empirical risk minimization* (ERM) over a function class on the training set. We denote this approach for choosing a loss function as *Target Metric Optimization* (TMO). This is especially more common with the advent of powerful general purpose function optimizers like deep networks and has also been rigorously analyzed for simpler function classes (Mohri et al., 2018).

Target Metric Optimization seems like a reasonable approach - if the practitioner knows about the target metric of interest for prediction using the model, it seems intuitive to optimize for the same objective on the training data. Prior work (both theoretical and applied) has both advocated for and argued against TMO for regression problems. Many prominent works on regression (Goldberger et al., 1964; Lecue & Mendelson, 2016) use the TMO approach, though most of them assume that the data is well behaved (e.g. sub-Gaussian noise). In terms of applications, many recent works on time-series forecasting (Wu et al., 2020; Oreshkin et al., 2019; Sen et al., 2019) also use the TMO approach directly on the target metric. On the other hand, the robust regression literature has long advocated for not using the target metric directly for ERM in the case of contamination or heavy tailed response/covariate behaviour (Huber, 1992; Hsu & Sabato, 2016; Zhu & Zhou, 2021; Lugosi & Mendelson, 2019a; Audibert et al., 2011; Brownlees et al., 2015) on account of its suboptimal

high-probability risk bounds. However, as noted in (Prasad et al., 2020), many of these methods are either not practical (Lugosi & Mendelson, 2019a; Brownlees et al., 2015) or have sub-optimal empirical performance (Hsu & Sabato, 2016). Even more practical methods such as (Prasad et al., 2020) would lead to sufficiently more computational overhead over standard TMO.

Another well known approach for designing a loss function is *Maximum Likelihood Estimation* (MLE). Here one assumes that the conditional distribution of $y$ given $x$ belongs to a family of distributions $p(y|\boldsymbol{x}; \boldsymbol{\theta})$ parameterized by $\boldsymbol{\theta} \in \Theta$ (McCullagh & Nelder, 2019). Then one can choose the negative log likelihood as the loss function to optimize using the training set, to obtain an estimate $\hat{\boldsymbol{\theta}}_{\text{mle}}$. This approach is sometimes used in the forecasting literature (Salinas et al., 2020; Davis & Wu, 2009) where the choice of a likelihood can encode prior knowledge about the data. For instance a negative binomial distribution can be used to model count data. During inference, given a new instance $\boldsymbol{x}'$, one can output the statistic from $p(y|\boldsymbol{x}'; \hat{\boldsymbol{\theta}}_{\text{mle}})$ that optimizes the target metric, as the prediction value (Gneiting, 2011). MLE also seems like a reasonable approach for loss function design - it is folklore that the MLE is asymptotically optimal for parameter estimation, in terms of having the smallest asymptotic variance among all estimators (Heyde, 1978; Rao, 1963), when the likelihood is well-specified. However, much less is known about finite-sample, fixed-dimension analysis of MLE, which is the typical regime of interest for the regression problems we consider in this paper. An important practical advantage for MLE is that model training is agnostic to the choice of the target metric - the same trained model can output estimators for different target metrics at inference time. Perhaps the biggest argument against the MLE approach is the requirement of knowing the likelihood distribution family. We address both these topics in Section 5.

Both TMO and MLE can be viewed as offering different approaches to selecting the loss function for a given regression model. In this paper, we argue that in several settings, both from a practical and theoretical perspective, MLE might be a better approach than TMO. This result might not be immediately obvious apriori - while MLE does benefit from prior knowledge of the distribution, TMO also benefits from prior knowledge of the target metric at training time.

Our *main contributions* are as follows:

**Competitiveness of MLE:** In Section 3, we prove that under some general conditions on the family of distributions and a property of interest, the MLE approach is competitive with any estimator for the property. We show that this result can be applied to fixed design regression problems in order to prove that MLE can be competitive (up to logarithmic terms) with any estimator in terms of excess square loss risk, under some assumptions.

**Example Applications:** In Section 4.1, we apply our general theorem to prove an excess square loss bound for the the MLE estimator for Poisson regression with the identity link (Nelder & Wedderburn, 1972; Lawless, 1987). We show that these bounds can be better than those of the TMO estimator, which in this case is least-squares regression. Then in Section 4.2, we show a similar application in the context of Pareto regression i.e $y|\boldsymbol{x}$ follows a Pareto distribution. We show that MLE can be competitive with robust estimators like the one in (Hsu & Sabato, 2016) and therefore can be better than TMO (least-squares).

**Empirical Results:** We propose the use of a general purpose mixture likelihood family (see Section 5) that can capture a wide variety of prior knowledge across datasets, including zero-inflated or bursty data, count data, sub-Gaussian continuous data as well as heavy tailed data, through different choices of (learnt) mixture weights. Then we empirically show that the MLE approach with this likelihood can outperform ERM for many different commonly used target metrics like WAPE, MAPE and RMSE (Hyndman & Koehler, 2006) for two popular forecasting and two regression datasets. Moreover the MLE approach is also shown to have better probabilistic forecasts (measured by quantile losses (Wen et al., 2017)) than quantile regression (Koenker & Bassett Jr, 1978; Gasthaus et al., 2019; Wen et al., 2017) which is the TMO approach in this case.

## 2 PRIOR WORK ON MLE

Maximum likelihood estimators (MLE) have been studied extensively in statistics starting with the work of Wald (1949); Redner (1981), who showed the maximum likelihood estimates are asymptotically consistent for parametric families. Fahrmeir & Kaufmann (1985) showed asymptotic

normality for MLE for generalized linear models. It is also known that under some regularity assumptions, MLE achieves the Cramer-Rao lower bound asymptotically (Lehmann & Casella, 2006). However, we note that none of these asymptotic results directly yield finite sample guarantees.

Finite sample guarantees have been shown for certain problem scenarios. Geer & van de Geer (2000); Zhang (2006) provided uniform convergence bounds in Hellinger distance for maximum likelihood estimation. These ideas were recently used by Foster & Krishnamurthy (2021) to provide algorithms for contextual bandits. There are other works which study MLE for non-parametric distributions e.g., Dümbgen & Rufibach (2009) showed convergence rates for log-concave distributions. There has been some works (Sur & Candès, 2019; Bean et al., 2013; Donoho & Montanari, 2016; El Karoui, 2018) that show that MLE can be sub-optimal for high dimensional regression i.e when the dimension grows with the number of samples. In this work we focus on the setting where the dimension does not scale with the number of samples.

Our MLE results differ from the above work as we provide finite sample competitiveness guarantees. Instead of showing that the maximum likelihood estimator converges in some distance metric, we show that under some mild assumptions it can work as well as other estimators. Hence, our methods are orthogonal to known well established results in statistics.

Perhaps the closest to our work is the competitiveness result of Acharya et al. (2017), who showed that MLE is competitive when the size of the output alphabet is bounded and applied to profile maximum likelihood estimation. In contrast, our work applies for unbounded output alphabets and can provide stronger guarantees in many scenarios.

## 3 COMPETITIVENESS OF MLE

In this section, we will show that under some reasonable assumptions on the likelihood family, the MLE is competitive with any estimator in terms of estimating any property of a distribution from the family. We will then show that this result can be applied to derive bounds on the MLE in some fixed design regression settings that can be better than that of TMO. We will first setup some notation.

**Notation:** Given a positive semi-definite symmetric matrix $M$, $\|x\|_M := x^T M x$ is the matrix norm of the vector $x$. $\lambda(M)$ denotes an eigenvalue of a symmetric square matrix $M$; specifically $\lambda_{\max}(M)$ and $\lambda_{\min}(M)$ denote the maximum and minimum eigenvalues respectively. The letter $f$ is used to denote general probability densities. We use $p$ to denote the conditional probability density of the response given the covariate. $\|\cdot\|_1$ will be overloaded to denote the $\ell_1$ norm between two probability distributions i.e $\|p - p'\|_1 := \int |p(z) - p'(z)|dz$. $D_{\mathrm{KL}}(p_1; p_2)$ will be used to denote the KL-divergence between the two distributions. If $\mathcal{Z}$ is a set equipped with a norm $\|\cdot\|$, then $\mathcal{N}(\epsilon, \mathcal{Z})$ will denote an $\epsilon$-net i.e any point $z \in \mathcal{Z}$ has a corresponding point $z' \in \mathcal{N}(\epsilon, \mathcal{Z})$ s.t $\|z - z'\| \leq \epsilon$. $\mathbb{B}_r^d$ denotes the ball centered at the origin with radius $r$ and $\mathbb{S}_r^{d-1}$ denotes its surface. We define $[n] := \{1, 2, \cdots, n\}$. $|\cdot|$ denotes the size of the enclosed set.

**General Competitiveness:** We first consider a general family of distributions $\mathcal{F}$ over the space $\mathcal{Z}$. For a sample $z \sim f$ (for $z \in \mathcal{Z}$ and $f \in \mathcal{F}$), the MLE distribution is defined as $f_z = \arg\max_{f \in \mathcal{F}} f(z)$. We are interested in estimating a property $\pi : \mathcal{F} \to \mathcal{W}$ of these distributions from an observed sample. The following definition will be required to impose some joint conditions on the distribution family and the property being estimated, that are needed for our result.

**Definition 1.** *The tuple $(\mathcal{F}, \pi)$, where $\mathcal{F}$ is a set of distributions and $\pi : \mathcal{F} \to \mathcal{W}$ a property of those distributions, is said to be $(T, \epsilon, \delta_1, \delta_2)$-approximable, if there exists a set of distributions $\tilde{\mathcal{F}} \subseteq \mathcal{F}$ s.t $|\tilde{\mathcal{F}}| \leq T$ and for every $f \in \mathcal{F}$, there exists a $\tilde{f}$ such that $\|f - \tilde{f}\|_1 \leq \delta_1$ and $\Pr_{z \sim f}\left(\left\|\pi(f_z) - \pi(\tilde{f}_z)\right\| \geq \epsilon\right) \leq \delta_2$, where $\tilde{f}_z = \arg\max_{\tilde{f} \in \tilde{\mathcal{F}}} \tilde{f}(z)$ and $\mathcal{W}$ has a norm $\|\cdot\|$.*

The above definition states that the set of distributions $\mathcal{F}$ has a finite $\delta$-cover, $\tilde{\mathcal{F}}$ in terms of the $\ell_1$ distance. Moreover the cover is such that solving MLE on the cover and applying the property $\pi$ on the result of the MLE is not too far from $\pi$ applied on the MLE over the whole set $\mathcal{F}$. This property is satisfied trivially if $\mathcal{F}$ is finite. We note that it is also satisfied by some commonly used set of distributions and corresponding properties. Now we state our main result.

**Theorem 1.** *Let $\hat{\pi}$ be an estimator such that for any $f \in \mathcal{F}$ and $z \sim f$, $\Pr(\|\pi(f) - \hat{\pi}(z)\| \geq \epsilon) \leq \delta$. Let $\mathcal{F}_f$ be a subset of $\mathcal{F}$ that contains $f$ such that with probability at least $1 - \delta$, $f_z \in \mathcal{F}_f$ and $(\mathcal{F}_f, \pi)$*

is $(T, \epsilon, \delta_1, \delta_2)$-approximable. Then the MLE satisfies the following bound,

$$\Pr(\|\pi(f) - \pi(f_z)\| \geq 3\epsilon) \leq (T+3)\delta + \delta_1 + \delta_2.$$

We provide the proof of Theorem 1 in Appendix A. We also provide a simpler version of this result for finite distribution families as Theorem 3 in Appendix A for the benefit of the reader.

**Competitiveness in Fixed Design Regression:** Theorem 1 can be used to show that MLE is competitive with respect to any estimator for square loss minimization in fixed design regression. We will first formally introduce the setting. Consider a fixed design matrix $\boldsymbol{X} \in \mathbb{R}^{n \times d}$ where $n$ is the number of samples and $d$ the feature dimension. We will work in a setting where $n \gg d$. The target vector is a random vector given by $y^n \in \mathbb{R}^n$. Let $y_i$ be the $i$-th coordinate of $y^n$ and $\boldsymbol{x}_i$ denote the $i$-th row of the design matrix. We assume that the target is generated from the conditional distribution given $\boldsymbol{x}_i$ such that,

$$y_i \sim p(\cdot | \boldsymbol{x}_i; \boldsymbol{\theta}^*), \ \ \boldsymbol{\theta}^* \in \Theta.$$

We are interested in optimizing a target metric $\ell(\cdot, \cdot)$ given an instance of the random vector $y^n$. The final objective is to optimize,

$$\min_{h \in \mathcal{H}} \mathbb{E}_{y_i \sim p(\cdot | \boldsymbol{x}_i; \boldsymbol{\theta}^*)} \left[ \frac{1}{n} \sum_{i=1}^n \ell(y_i, h(\boldsymbol{x}_i)) \right],$$

where $\mathcal{H}$ is a class of functions. In this context, we are interested in comparing two approaches.

**TMO** (see (Mohri et al., 2018)). This is standard empirical risk minimization on the target metric where given an instance of the random vector $y^n$ one outputs the estimator $\hat{h} = \min_{h \in \mathcal{H}} \frac{1}{n} \sum_{i=1}^n \ell(y_i, h(\boldsymbol{x}_i))$.

**MLE and post-hoc inference** (see (Gneiting, 2011)). In this method one first solves for the parameter in the distribution family that best explains the empirical data by MLE i.e.,

$$\hat{\boldsymbol{\theta}}_{\text{mle}} := \underset{\boldsymbol{\theta} \in \Theta}{\operatorname{argmin}} \mathcal{L}(y^n; \theta), \ \ \text{where } \mathcal{L}(y^n; \theta) := \sum_{i=1}^n -\log p(y_i | \boldsymbol{x}_i; \boldsymbol{\theta})$$

Then during inference given a sample $\boldsymbol{x}_i$ the predictor is defined as, $\tilde{h}(\boldsymbol{x}_i) := \operatorname{argmin}_{\hat{y}} \mathbb{E}_{y \in p(\cdot | \boldsymbol{x}_i; \hat{\boldsymbol{\theta}}_{\text{mle}})} [\ell(y, \hat{y})]$ or in other words we output the statistic from the MLE distribution that optimizes the loss function of interest. For instance if $\ell$ is the square loss, then $\tilde{h}(\boldsymbol{x}_i)$ will be the mean of the conditional distribution $p(y | \boldsymbol{x}_i; \hat{\boldsymbol{\theta}}_{\text{mle}})$.

We will prove a general result using Theorem 1 when the target metric $\ell$ is the square loss and $\mathcal{H}$ is a linear function class. Moreover, the true distribution $p(\cdot | \boldsymbol{x}_i; \boldsymbol{\theta}^*)$ is such that $\mathbb{E}[y_i] = \langle \boldsymbol{\theta}^*, \boldsymbol{x}_i \rangle$ for all $i \in [n]$ i.e we are in the linear realizable setting.

In this case the quantity of interest is the excess square loss risk given by,

$$\mathcal{E}(\boldsymbol{\theta}) := \frac{1}{n} \sum_{i=1}^n \mathbb{E}_{y^n} \|y^n - \boldsymbol{X}\boldsymbol{\theta}\|_2^2 - \frac{1}{n} \sum_{i=1}^n \mathbb{E}_{y^n} \|y^n - \boldsymbol{X}\boldsymbol{\theta}^*\|_2^2 = \|\boldsymbol{\theta} - \boldsymbol{\theta}^*\|_\Sigma^2, \tag{1}$$

where $\Sigma := (\sum_i \boldsymbol{x}_i \boldsymbol{x}_i^T)/n$ is the normalized covariance matrix, and $\boldsymbol{\theta}^*$ is the population minimizer of the target metric over the linear function class. Now we are ready to state the main result.

**Theorem 2.** *Consider a fixed design regression setting where the likelihood family is parameterized by $\boldsymbol{\theta} \in \Theta \subseteq \mathbb{B}_w^d$ and $|\mathcal{N}(\epsilon, \Theta \cap \mathbb{B}_w^d)| \leq |\mathcal{N}(\epsilon, \mathbb{B}_w^d)|$ for a small enough $\epsilon$. Further the following conditions hold,*

1. $D_{\text{KL}}(p(y_i; \boldsymbol{\theta}); p(y_i; \boldsymbol{\theta}')) \leq L\|\boldsymbol{\theta} - \boldsymbol{\theta}'\|_2$.
2. *The negative log-likelihood $\mathcal{L}(y^n; \boldsymbol{\theta})$ as a function of $\boldsymbol{\theta}$ is $\alpha$-strongly convex and $\beta$-smooth, w.p. at least $1 - \delta$.*

*Further suppose there exists an estimator $\boldsymbol{\theta}_{\text{est}}$ such that $\mathcal{E}(\boldsymbol{\theta}_{\text{est}}) \leq (c_1 + c_2 \log(1/\delta))^\eta / n$, where $c_1, c_2$ are problem dependent quantities and $\eta > 0$. Then the MLE estimator also satisfies,*

$$\mathcal{E}(\hat{\boldsymbol{\theta}}_{\text{mle}}) = O\left(\frac{\left(c_1 + c_2 d\left(\log n + \log(wL\lambda_{\max}(\Sigma)) + \log(\beta/\alpha) + \log \frac{1}{\delta}\right)\right)^{\eta}}{n}\right)$$

*w.p at least* $1 - \delta$.

We provide the proof in Appendix C. The proof involves proving the conditions in Theorem 1 and bounding the size of the cover $T$.

In order to better understand Theorem 2, let us consider a typical case where there exists a possibly complicated estimator such that $\mathcal{E}(\boldsymbol{\theta}_{\text{est}}) = O((d + \log(1/\delta))/n)$. In this case the above theorem implies that MLE will be competitive with this estimator up to a $\log n$ factor. In many cases the MLE might be much simpler to implement than the original estimator but would essentially match the same error bound. We now provide specific examples in subsequent sections.

## 4 APPLICATIONS OF COMPETITIVENESS RESULT

In this section we will specialize to two examples, Poisson regression and Pareto regression, where we show that MLE can be better than TMO through the use of our competitive result in Theorem 2.

### 4.1 POISSON REGRESSION

We work in the fixed design setting in Section 3 and assume that the conditional distribution of $y|\boldsymbol{x}$ is Poisson i.e,

$$p(y_i = k | \boldsymbol{x}_i; \boldsymbol{\theta}^*) = \frac{\mu_i^k e^{-\mu_i}}{k!} \quad \text{where,} \quad \mu_i = \langle \boldsymbol{\theta}^*, \boldsymbol{x}_i \rangle > 0, \tag{2}$$

for all $i \in [n]$. Poisson regression is a popular model for studying count data regression which naturally appears in many applications like demand forecasting (Lawless, 1987). Note that here we study the version of Poisson regression with the identity link function (Nelder & Wedderburn, 1972), while another popular variant is the one with exponential link function (McCullagh & Nelder, 2019). We choose the identity link function for a fair comparison of the two approaches as it is realizable for both the approaches under the linear function class i.e the globally optimal estimator in terms of population can be obtained by both approaches. The exponential link function would make the problem non-realizable under a linear function class for the TMO approach.

We make the following natural assumptions. Let $\Sigma = (\sum_{i=1}^{n} \boldsymbol{x}_i \boldsymbol{x}_i^T)/n$ be the design covariance matrix as before and $\boldsymbol{M} = (\sum_{i=1}^{n} \mu_i \boldsymbol{u}_i \boldsymbol{u}_i^T)/n$, where $\boldsymbol{u}_i = \boldsymbol{x}_i/\|\boldsymbol{x}_i\|_2$. Let $\chi$ and $\zeta$ be the condition numbers of the matrices $\boldsymbol{M}$ and $\Sigma$ respectively.

**Assumption 1.** *The parameter space $\Theta$ and the design matrix $\boldsymbol{X}$ satisfy the following,*

- *(A1) The parameter space $\Theta = \{\boldsymbol{\theta} \in \mathbb{R}^d : \|\boldsymbol{\theta}\|_2 \leq w, \min(\|\boldsymbol{\theta}\|_2^2, \langle \boldsymbol{\theta}, \boldsymbol{x}_i \rangle) \geq \gamma > 0, \ \forall i \in [n]\}$.*
- *(A2) The design matrix is such that $\lambda_{\min}(\Sigma) > 0$ and $\|\boldsymbol{x}_i\|_2 \leq R$ for all $i \in [n]$.*
- *(A3) Let $\lambda_{\min}(\boldsymbol{M}) \geq \frac{R^2}{4n\gamma^2}(d\log(24\chi) + \log(1/\delta))$ and $\sqrt{\lambda_{\max}(\boldsymbol{M})(d\log(24\chi) + \log(1/\delta))} \leq \sqrt{n}\lambda_{\min}(\boldsymbol{M})/16$, for a small $\delta \in (0,1)$ [1].*

The above assumptions are fairly mild. For instance $\lambda_{\min}$ is $\tilde{\Omega}(1/d)$ for random Gaussian covariance matrices (Bai & Yin, 2008). The other part merely requires that $\lambda_{\min}(\boldsymbol{M}) = \tilde{\Omega}(\sqrt{d\lambda_{\max}(\boldsymbol{M})/n})$.

We are interested in comparing MLE with TMO for the square loss which is just the least-squares estimator i.e $\hat{\boldsymbol{\theta}}_{\text{ls}} := \operatorname{argmin}_{\boldsymbol{\theta} \in \Theta} \frac{1}{n}\|y^n - \boldsymbol{X}\boldsymbol{\theta}\|_2^2$. Note that it is apriori unclear as to which approach would be better in terms of the target metric because on one hand the MLE method knows the distribution family but on the other hand TMO is explicitly geared towards minimizing the square loss during training.

Least squares analysis is typically provided for regression under sub-Gaussian noise. By adapting existing techniques (Hsu et al., 2012), we show the following guarantee for Poisson regression with least square loss. We provide a proof in Appendix D for completeness.

---

[1] Note that the constants can be further tightened in our analysis.

**Lemma 1.** *Let $\mu_{max} = \max_i \mu_i$. The least squares estimator $\hat{\boldsymbol{\theta}}_{\mathrm{ls}}$ satisfies the following loss bounds w.p. at least $1 - \delta$,*

$$\mathcal{E}(\hat{\boldsymbol{\theta}}_{\mathrm{ls}}) = \begin{cases} O\big(\frac{\mu_{max}}{n}\big(\log\frac{1}{\delta} + d\big)\big) & \textit{if } \mu_{max} \geq (\log(1/\delta) + d\log 6)/2 \\ O\big(\frac{1}{n}\big(\log\frac{1}{\delta} + d\big)^2\big) & \textit{otherwise} \end{cases}$$

Now we present our main result in this section which uses the competitiveness bound in Theorem 2 coupled with the existence of a superior estimator compared to TMO, to show that the MLE estimator can have a better bound than TMO.

In Theorem 4 (in Appendix F), under some mild assumptions on the covariates, we construct an estimator $\boldsymbol{\theta}_{\mathrm{est}}$ with the following bound for the Poisson regression setting,

$$\mathcal{E}(\boldsymbol{\theta}_{\mathrm{est}}) \leq c \cdot \|\boldsymbol{\theta}^*\|^2 \cdot \lambda_{\max}(\Sigma)\left(\frac{d + \log(\frac{1}{\delta})}{n}\right). \tag{3}$$

The construction of the estimator is median-of-means tournament based along the lines of (Lugosi & Mendelson, 2019a) and therefore the estimator might not be practical. However, this immediately gives the following bound on the MLE as a corollary of Theorem 2.

**Corollary 1.** *Under assumption 1 and the conditions of Theorem 4, the MLE estimator for the Poisson regression setting satisfies w.p. at least $1 - \delta$,*

$$\mathcal{E}(\hat{\boldsymbol{\theta}}_{\mathrm{mle}}) = O\left(\|\boldsymbol{\theta}^*\|^2 \cdot \lambda_{\max}(\Sigma)\frac{d(\log n + \log(wR\lambda_{\max}(\Sigma)) + \log\chi + \log\frac{1}{\delta})}{n}\right).$$

The bound in Corollary 1 can be better than the bound for $\hat{\boldsymbol{\theta}}_{\mathrm{ls}}$ in Lemma 1. In the sub-Gaussian region, the bound in Lemma 1 scales linearly with $\mu_{max}$ which can be prohibitively large even when a few covariates have large norms. The bound for the MLE estimator in Corollary 1 has no such dependence. Further, in the sub-Exponential region the bound in Lemma 1 scales as $\tilde{O}(d^2/n)$ while the bound in Corollary 1 has a $\tilde{O}(d/n)$ dependency, up to log-factors. In Appendix G, we also show that when the covariates are one-dimensional, an even sharper analysis is possible, that shows that the MLE estimator is always better than least squares in terms of excess risk. In Appendix I.8, we perform a simulated experiment that shows a linear growth of the test error w.r.t $\lambda_{\max}(\Sigma)$, further showing the efficacy of our theoretical bounds.

## 4.2 PARETO REGRESSION

Now we will provide an example of a heavy tailed regression setting where it is well-known that TMO for the square loss does not perform well (Lugosi & Mendelson, 2019a). We will consider the Pareto regression setting given by,

$$p(y_i|\boldsymbol{x}_i) = \frac{bm_i^b}{y_i^{b+1}}, \quad m_i = \frac{b-1}{b}\langle\boldsymbol{\theta}^*, \boldsymbol{x}_i\rangle \quad \text{for } y_i \geq m_i \tag{4}$$

provided $\langle\boldsymbol{\theta}^*, \boldsymbol{x}_i\rangle > \gamma$ for all $i \in [n]$. Thus $y_i$ is Pareto given $\boldsymbol{x}_i$ and $\mathbb{E}[y_i|\boldsymbol{x}_i] = \mu_i := \langle\boldsymbol{\theta}^*, \boldsymbol{x}_i\rangle$. We will assume that $b > 4$ such that $4 + \epsilon$-moment exists for $\epsilon > 0$. As in the Poisson setting, we choose this parametrization for a fair comparison between TMO and MLE i.e in the limit of infinite samples $\boldsymbol{\theta}^*$ lies in the linear solution space for both TMO (least squares) and MLE.

As before, to apply Theorem 2 we need an estimator with a good risk bound. We use the estimator in Theorem 4 of (Hsu & Sabato, 2016), which in the fixed design pareto regression setting yields,

$$\mathcal{E}(\boldsymbol{\theta}_{\mathrm{est}}) = \left(1 + O\left(\frac{d\log\frac{1}{\delta}}{n}\right)\right)\frac{\|\theta^*\|_\Sigma^2}{b(b-2)}.$$

Note that the above estimator might not be easily implementable, however this yields the following corollary of Theorem 2, which is a bound on the performance of the MLE estimator.

**Corollary 2.** *Under assumptions of our Pareto regression setting, the MLE estimator satisfies w.p. at least $1 - \delta$,*

$$\mathcal{E}(\hat{\boldsymbol{\theta}}_{\mathrm{mle}}) = 1 + O\left(\frac{d^2\left(\log n + \log\zeta + \log\frac{bwR\lambda_{\max}(\Sigma)}{\gamma} + \log\frac{1}{\delta}\right)}{n}\right)\frac{\|\theta^*\|_\Sigma^2}{b(b-2)}.$$

The proof is provided in Appendix H. It involves verifying the two conditions in Theorem 2 in the Pareto regression setting.

The above MLE guarantee is expected to be much better than what can be achieved by TMO which is least-squares. It is well established in the literature (Hsu & Sabato, 2016; Lugosi & Mendelson, 2019a) that ERM on square loss cannot achieve a $O(\log(1/\delta))$ dependency in a heavy tailed regime; instead it can achieve only a $O(\sqrt{1/\delta})$ rate.

## 5 Choice of Likelihood and Inference Methods

In this section we discuss some practical considerations for MLE, such as adapting to a target metric of interest at inference time, and the choice of the likelihood family.

**Inference for different target metrics:**  In most practical settings, the trained regression model is used at inference time to predict the response variable on some test set to minimize some target metric. For the MLE based approach, once the distribution parameters are learnt, this involves using an appropriate statistic of the learnt distribution at inference time (see Section 3). For mean squared error and mean absolute error, the estimator corresponds to the mean and median of the distribution, but for several other commonly used loss metrics in the forecasting domain such as Mean Absolute Percentage Error (MAPE) and Relative Error (RE) (Gneiting, 2011; Hyndman & Koehler, 2006), this estimator corresponds to a median of a transformed distribution (Gneiting, 2011). Please see Appendix I for more details. This ability to optimize the estimator at inference time for different target metrics using a *single trained model* is another advantage that MLE based approaches have over TMO models that are trained individually for specific target metrics.

**Mixture Likelihood:**  An important practical question when performing MLE-based regression is to decide which distribution family to use for the response variable. The goal is to pick a distribution family that can capture the inductive biases present in the data. It is well known that a misspecified distribution family for MLE might adversely affect generalization error of regression models (Heagerty & Kurland, 2001). At the same time, it is also desirable for the distribution family to be generic enough to cater to diverse datasets with potentially different types of inductive biases, or even datasets for which no distributional assumptions can be made in advance.

A simple approach that we observe to work particularly well in practice with regression models using deep neural networks is to assume the response variable comes from a mixture distribution, where each mixture component belongs to a different distribution family and the mixture weights are learnt along with the parameters of the distribution. More specifically, we consider a mixture distribution of $k$ components $p(y|\boldsymbol{x}; \boldsymbol{\theta}_1, \ldots, \boldsymbol{\theta}_k, w_1, \ldots, w_k) = \sum_{j=1}^{k} w_j p_j(y|\boldsymbol{x}; \boldsymbol{\theta}_j)$, where each $p_j$ characterizes a different distribution family, and the mixture weights $w_j$ and distribution parameters $\boldsymbol{\theta}_j$ are learnt together. We would like to have a mixture distribution that can handle different situations like sparse data, sub-Exponential and sub-Gaussian tails, count data as well as heavy tailed data. Moreover it should be applicable to continuous valued datasets in general.

We use a three component mixture of $(i)$ the constant 0 (zero-inflation for dealing with bi-modal sparse data), $(ii)$ a continuous version of negative binomial where $n$ and $p$ are learnt and $(iii)$ a Pareto distribution where the scale parameter is learnt. We provide more details about the continuous version of negative binomial distribution in Appendix I.2. Our experiments in Section 6 show that this mixture shows promising performance on a variety of datasets.

This will increase the number of parameters and the resulting likelihood might require non-convex optimization. However, we empirically observed that with sufficiently over-parameterized networks and gradient-based optimizers, this is not a problem in practice (Fig. 4 shows a convergence curve).

## 6 Empirical Results

We present empirical results on two time-series forecasting problems and two regression problems using neural networks. We will first describe our models and baselines. Our goal is to compare the MLE approach with the TMO approach for three target metrics per dataset.

| Model | Favorita | | | M5 | | |
|---|---|---|---|---|---|---|
| | MAPE | WAPE | RMSE | MAPE | WAPE | RMSE |
| TMO(MSE) | 0.6121±0.0075 | 0.2891±0.0023 | 175.3782±0.8235 | 0.5045±0.004 | 0.2839±0.0008 | 7.507±0.023 |
| TMO(MAE) | 0.3983±0.0012 | 0.2258±0.0006 | **161.4919**±0.4748 | 0.4452±0.0005 | **0.266**±0.0001 | **7.0503**±0.0094 |
| TMO(MAPE) | 0.3199±0.0011 | 0.2528±0.0016 | 192.3823±1.3871 | 0.3892±0.0001 | 0.3143±0.0007 | 11.3799±0.1965 |
| TMO(Huber) | 0.432±0.0033 | 0.2366±0.0018 | 164.7006±0.7178 | 0.4722±0.0007 | 0.269±0.0002 | 7.093±0.0133 |
| MLE(ZNBP) | **0.3139**±0.0011 | **0.2238**±0.0009 | 164.6521±1.5185 | **0.3864**±0.0001 | 0.2677±0.0002 | 7.2133±0.0152 |

Table 1: We provide the MAPE, WAPE and RMSE metrics for all the models on the test set of two time-series datasets. The confidence intervals provided are one standard error over 50 experiments, for each entry. TMO(<loss>) refers to TMO using the <loss>. For the MLE row, we only train one model per dataset. The same model is used to output a different statistic for each column during inference. For MAPE, we output the optimizer of MAPE given in Section I.5. For WAPE we output the median and for RMSE we output the mean.

| Model | Bicycle Share | | | Gas Turbine | | |
|---|---|---|---|---|---|---|
| | MAPE | WAPE | RMSE | MAPE | WAPE | RMSE |
| TMO(MSE) | 0.2503±0.0008 | 0.1421±0.0003 | 878.5815±1.3059 | 0.8884±0.0118 | 0.3496±0.0041 | 1.5628±0.0071 |
| TMO(MAE) | 0.2594±0.0011 | 0.1436±0.0003 | 901.1357±1.4943 | **0.774**±0.0054 | **0.3389**±0.0019 | 1.5789±0.0067 |
| TMO(MAPE) | 0.2382±0.0012 | 0.1469±0.0012 | 899.9163±4.8219 | 0.8108±0.0009 | 0.8189±0.001 | 3.0573±0.0019 |
| TMO(Huber) | 0.2536±0.0011 | 0.1414±0.0004 | 889.1173±1.9654 | 0.902±0.0128 | 0.3598±0.0049 | 1.5992±0.0082 |
| MLE(ZNBP) | **0.1969**±0.0018 | **0.1235**±0.001 | **767.4368**±7.1274 | 0.9877±0.0019 | 0.3379±0.0004 | **1.4547**±0.0054 |

Table 2: We provide the MAPE, WAPE and RMSE metrics for all the models on the test set of two regression datasets. The confidence intervals provided are one standard error over 50 experiments, for each entry. TMO(<loss>) refers to TMO using the <loss>. For the MLE row, we only train one model per dataset. The same model is used to output a different statistic for each column during inference. For MAPE, we output the optimizer of MAPE given in Section I.5. For WAPE we output the median and for RMSE we output the mean.

**Common Experimental Protocol:** Now we describe the common experimental protocol on all the datasets (we get into dataset related specifics and architectures subsequently). For a fair comparison the architecture is kept the same for TMO and MLE approaches. For each dataset, we tune the hyper-parameters for the TMO(MSE) objective. Then these hyper-parameters are held fixed for all models for that dataset i.e only the output layer and the loss function is modified. We provide all the details in Appendix I. Our code will be available here.

For the MLE approach, the output layer of the models map to the MLE parameters of the mixture distribution introduced in Section 5, through link functions. The MLE output has 6 parameters, three for mixture weights, two for negative binomial component and one for the scale parameter in Pareto. The choice of the link functions and more details are specified in Appendix I.3. The loss function used is the negative log-likelihood implemented in Tensorflow Probability (Dillon et al., 2017). Note that for the MLE approach *only one* model is trained per dataset and during inference we output the statistic that optimizes the target metric in question. We refer to our MLE based models that employ the mixture likelihood from Section 5 as MLE(ZNBP) loss, where ZNBP refers to the mixture components **Z**ero, **N**egative-**B**inomial and **P**areto.

For TMO, the output layer of the models map to $\hat{y}$ and we directly minimize the target metric in question. Note that this means we need to train *a separate model for every target metric*. Thus we have one model each for target metrics in {'MSE', 'MAE', 'MAPE'}. Further we also train a model using the Huber loss [2]. In order to keep the number of parameters the same as that of MLE, we add an additional 6 neurons to the TMO models.

## 6.1 Experiments on Forecasting Datasets

We perform our experiments on two well-known forecasting datasets used in Kaggle competitions.

1. The M5 dataset (M5, 2020) consists of time series data of product sales from 10 Walmart stores in three US states. The data consists of two different hierarchies: the product hierarchy and store location hierarchy. For simplicity, in our experiments we use only the product hierarchy consisting of 3K individual time-series and 1.8K time steps.
2. The Favorita dataset (Favorita, 2017) is a similar dataset, consisting of time series data from Corporación Favorita, a South-American grocery store chain. As above, we use the product hierarchy, consisting of 4.5k individual time-series and 1.7k time steps.

The task is to predict the values for the last 14 days all at once. The preceding 14 days are used for validation. We provide more details about the dataset generation for reproducibility in Appendix I.

---

[2]The Huber loss is commonly used in robust regression (Huber, 1992; Lugosi & Mendelson, 2019a)

| Model | p10QL | p90QL |
|---|---|---|
| TMO (Quantile) | 0.0973±0.0002 | 0.0628±0.0019 |
| MLE(ZNBP) | **0.0788**±0.0008 | **0.0536**±0.0007 |

Table 3: The MLE model predicts the empirical quantile of interest during inference. It is compared with Quantile regression (TMO based). The results, averaged over 50 runs along with the corresponding confidence intervals are presented.

| Model | MAPE | WAPE | RMSE |
|---|---|---|---|
| MLE(NB) | 0.3314+/-0.0016 | 0.2521+/-0.002 | 175.501+/-1.1928 |
| MLE(ZNB) | 0.3186+/-0.0011 | 0.2453+/-0.002 | 170.0075+/-1.282 |
| MLE(ZNBP) | **0.3139**±0.0011 | **0.2238**±0.0009 | **164.6521**+/-1.5185 |

Table 4: We perform an ablation study on the Favorita dataset, where we progressively add the components of our mixture distribution. There are three MLE models in the progression: Negative Binomial (NB), Zero-Inflated Negative Binomial (ZNB) and finally ZNBP.

The base architecture for the baselines as well as our model is a seq-2-seq model (Sutskever et al., 2014). The encoders and decoders both are LSTM cells (Hochreiter & Schmidhuber, 1997). The architecture is illustrated in Figure 1 and described in more detail in Appendix I.

We present our experimental results in Table 1. On both the datasets the MLE model with the appropriate inference-time estimator for a metric is always better than TMO trained on the same target metric, except for WAPE in M5 where MLE's performance is only marginally worse. Note that the MLE model is always the best or second best performing model on *all* metrics, among *all* TMO models. For TMO the best performance is not always achieved for the same target metric. For instance, TMO(MAE) performs better than TMO(MSE) for the RMSE metric on the Favorita dataset. In Table 4 we perform an ablation study on the Favorita dataset, where we progressively add mixture components resulting in three MLE models: Negative Binomial, Zero-Inflated Negative Binomial and finally ZNBP. This shows that each of the components add value in this dataset.

## 6.2 EXPERIMENTS ON REGRESSION DATASETS

We perform our experiments on two standard regression datasets,

1. The Bicyle Share dataset (Bicycle, 2017) has daily counts of the total number of rental bikes. The features include time features as well as weather conditions such as temperature, humidity, windspeed etc. A random 10% of the dataset is used as test and the rest for training and validation. The dataset has a total of 730 samples.
2. The Gas Turbine dataset (Kaya et al., 2019) has 11 sensor measurements per example (hourly) from a gas turbine in Turkey. We consider the level of NOx as our target variable and the rest as predictors. There are 36733 samples in total. We use the official train/test split. A randomly chosen 20% of the training set is used for validation. The response variable is continuous.

For all our models, the model architecture is a fully connected DNN with one hidden layer that has 32 neurons. Note that for categorical variables, the input is first passed through an embedding layer (one embedding layer per feature), that is jointly trained. We provide further details like the shape of the embedding layers in Appendix I. The architecture is illustrated in Figure 2.

We present our results in Table 2. On the Bicycle Share dataset, the MLE(ZNBP) based model performs optimally in all metrics and often outperforms the TMO models by a large margin even though TMO is a separate model per target metric. On the Gas Turbine dataset, the MLE based model is optimal for WAPE and RMSE, however it does not perform that well for the MAPE metric.

In Table 3, we compare the MLE based approach versus quantile regression (TMO based) on the Bicycle Share dataset, where the metric presented is the normalized quantile loss (Wang et al., 2019). We train the TMO model for the corresponding quantile loss directly and the predictions are evaluated on normalized quantile losses as shown in the table. The MLE based model is trained by minimizing the negative log-likelihood and then during inference we output the corresponding empirical quantile from the predicted distribution. MLE(ZNBP) outperforms TMO(Quantile) significantly.

**Discussion:** We compare the approaches of direct ERM on the target metric (TMO) and MLE followed by post-hoc inference time optimization for regression and forecasting problems. We prove a general competitiveness result for the MLE approach and also show theoretically that it can be better than TMO in the Poisson and Pareto regression settings. Our empirical results show that our proposed general purpose likelihood function employed in the MLE approach can uniformly perform well on several tasks across four datasets. Even though this addresses some of the concerns about choosing the correct likelihood for a dataset, some limitations still remain for example concerns about the non-convexity of the log-likelihood. We provide a more in-depth discussion in Appendix J.

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

## A    PROOF FOR COMPETITIVENESS OF MLE

**Theorem 3.** *Let $\hat{\pi}$ be an estimator for a property $\pi$, such that for any $f \in \mathcal{F}$ and $z \sim f$, $\Pr(\|\pi(f) - \hat{\pi}(z)\| \geq \epsilon) \leq \delta$. Then the MLE based estimator satisfies the following bound,*

$$\Pr(\|\pi(f) - \pi(f_z)\| \geq 2\epsilon) \leq (|\mathcal{F}| + 1)\delta.$$

*Proof of Theorem 3.* By triangle inequality and by the properties,

$$\|\pi(f) - \pi(f_z)\| \leq \|\pi(f) - \hat{\pi}(z)\| + \|\hat{\pi}(z) - \pi(f_z)\|.$$

Hence,

$$1_{\|\pi(f) - \pi(f_z)\| \geq 2\epsilon} \leq 1_{\|\pi(f) - \hat{\pi}(z)\| \geq \epsilon} + 1_{\|\hat{\pi}(z) - \pi(f_z)\| \geq \epsilon}.$$

We now take expectation of both LHS and RHS of the above equation with respect to the distribution $f$. For the LHS, observe that

$$\mathbb{E}[1_{\|\pi(f) - \pi(f_z)\| \geq 2\epsilon}] = \Pr(\|\pi(f) - \pi(f_z)\| \geq 2\epsilon).$$

For the first term in the RHS

$$\mathbb{E}[1_{\|\pi(f) - \hat{\pi}(z)\| \geq \epsilon}] \leq \Pr(\|\pi(f) - \hat{\pi}(z)\| \geq \epsilon) \leq \delta,$$

by the assumption in the theorem. Combining the above three equations yield that

$$\Pr(\|\pi(f) - \pi(f_z)\| \geq 2\epsilon) \leq \mathbb{E}[1_{\|\hat{\pi}(z) - \pi(f_z)\| \geq \epsilon}] + \delta.$$

We now prove that $\mathbb{E}[1_{\|\hat{\pi}(z) - \pi(\tilde{f}_z)\| \geq \epsilon}] \leq |\mathcal{F}|\delta.$

$$\begin{aligned}
\mathbb{E}[1_{\|\hat{\pi}(z) - \pi(f_z)\| \geq \epsilon}] &= \int_z f(z) 1_{\|\hat{\pi}(z) - \pi(f_z)\| \geq \epsilon} \\
&\overset{(a)}{\leq} \int_z f_z(z) 1_{\|\hat{\pi}(z) - \pi(f_z)\| \geq \epsilon} \\
&\leq \int_z \sum_f f(z) 1_{\|\hat{\pi}(z) - \pi(f)\| \geq \epsilon} \\
&\overset{(b)}{=} \sum_f \int_z f(z) 1_{\|\hat{\pi}(z) - \pi(f)\| \geq \epsilon} \\
&= \sum_f \Pr_{z \sim f}(\|\hat{\pi}(z) - \pi(f)\| \geq \epsilon) \\
&\leq \sum_f \delta = |\mathcal{F}|\delta,
\end{aligned}$$

where $(a)$ uses the fact that $f_z$ is the MLE estimate over set $\mathcal{F}$ and hence $f_z(z) \geq f(z)$. $(b)$ follows from Fubini's theorem.

$\square$

Now we prove an extension of our last theorem that can deal with infinite likelihood families.

*Proof of Theorem 1.* Let $\tilde{\mathcal{F}}$ be the cover of $\mathcal{F}_f$ that satisfies assumptions in Definition 1. By triangle inequality and by the properties of $\tilde{\mathcal{F}}$, with probability at least $1 - \delta_2 - \delta$,

$$\begin{aligned}
\|\pi(f) - \pi(f_z)\| &\leq \|\pi(f) - \hat{\pi}(z)\| + \|\hat{\pi}(z) - \pi(\tilde{f}_z)\| + \|\pi(\tilde{f}_z) - \pi(f_z)\| \\
&\leq \|\pi(f) - \hat{\pi}(z)\| + \|\hat{\pi}(z) - \pi(\tilde{f}_z)\| + \epsilon
\end{aligned}$$

We have used the fact that $f_z \in \mathcal{F}_f$ w.p at least $1 - \delta$. Hence with probability at least $1 - \delta_2 - \delta$,

$$1_{\|\pi(f) - \pi(f_z)\| \geq 3\epsilon} \leq 1_{\|\pi(f) - \hat{\pi}(z)\| \geq \epsilon} + 1_{\|\hat{\pi}(z) - \pi(\tilde{f}_z)\| \geq \epsilon}.$$

We now take expectation of both LHS and RHS of the above equation with respect to the distribution $f$. For the LHS, observe that

$$\mathbb{E}[1_{\|\pi(f)-\pi(f_z)\|\geq 3\epsilon}] = \Pr(\|\pi(f) - \pi(f_z)\| \geq 3\epsilon).$$

For the first term in the RHS

$$\mathbb{E}[1_{\|\pi(f)-\hat{\pi}(z)\|\geq\epsilon}] \leq \Pr(\|\pi(f) - \hat{\pi}(z)\| \geq \epsilon) \leq \delta,$$

by the assumption in the theorem. Combining the above three equations yield that

$$\Pr(\|\pi(f) - \pi(f_z)\| \geq 3\epsilon) \leq \mathbb{E}[1_{\|\hat{\pi}(z)-\pi(\tilde{f}_z)\|\geq\epsilon}] + 2\delta + \delta_2.$$

We now prove that $\mathbb{E}[1_{\|\hat{\pi}(z)-\pi(\tilde{f}_z)\|\geq\epsilon}] \leq T\delta + \delta_1$. Let $\tilde{f}$ be the distribution in $\tilde{\mathcal{F}}_f$ that is at most $\delta_1$ away from $f$. Then,

$$
\begin{aligned}
\mathbb{E}[1_{\|\hat{\pi}(z)-\pi(\tilde{f}_z)\|\geq\epsilon}] &= \int_z f(z)1_{\|\hat{\pi}(z)-\pi(\tilde{f}_z)\|\geq\epsilon} \\
&\stackrel{(a)}{\leq} \int_z \tilde{f}(z)1_{\|\hat{\pi}(z)-\pi(\tilde{f}_z)\|\geq\epsilon} + \|f - \tilde{f}\|_1 \\
&\stackrel{(b)}{\leq} \int_z \tilde{f}(z)1_{\|\hat{\pi}(z)-\pi(\tilde{f}_z)\|\geq\epsilon} + \delta_1 \\
&\stackrel{(c)}{\leq} \int_z \tilde{f}_z(z)1_{\|\hat{\pi}(z)-\pi(\tilde{f}_z)\|\geq\epsilon} + \delta_1 \\
&\leq \int_z \sum_{\tilde{f}\in\tilde{\mathcal{F}}} \tilde{f}(z)1_{\|\hat{\pi}(z)-\pi(\tilde{f}_z)\|\geq\epsilon} + \delta_1 \\
&\stackrel{(d)}{=} \sum_{\tilde{f}\in\tilde{\mathcal{F}}} \int_z \tilde{f}(z)1_{\|\hat{\pi}(z)-\pi(\tilde{f})\|\geq\epsilon} + \delta_1 \\
&= \sum_{\tilde{f}\in\tilde{\mathcal{F}}} \Pr_{z\sim\tilde{f}}(\|\pi(\tilde{f}) - \hat{\pi}(z)\| \geq \epsilon) + \delta_1 \\
&\leq \left(\sum_{\tilde{f}\in\tilde{\mathcal{F}}} \delta\right) + \delta_1 = T\delta + \delta_1,
\end{aligned}
$$

where the $(a)$ follows from the definition of $\ell_1$ distance between distributions, $(b)$ follows from the properties of the cover, $(c)$ uses the fact that $\tilde{f}_z$ is the MLE estimate over set $\tilde{\mathcal{F}}$ and hence $\tilde{f}_z(z) \geq \tilde{f}(z)$. $(d)$ follows from Fubini's Theorem.

$\square$

## B    OTHER GENERAL RESULTS FOR THE MLE

In this section, we prove that the log likelihood of the MLE distribution at the observed data point is is close to the log-likelihood of the ground truth at the data point, upto an additive factor that depends on the Shtarkov sum of the family.

**Lemma 2.** *Let $z \sim f$. The maximum likelihood estimator satisfies the following inequality with probability at least $1 - \delta$,*

$$\log f(z) \leq \log f_z(z) \leq \log f(z) + \log\left(\sum_{z\in\mathcal{Z}} f_z(z)\right) + \log\frac{1}{\delta},$$

*where $\sum_{z\in\mathcal{Z}} f_z(z)$ is the Shtarkov sum of the family $\mathcal{F}$. Furthermore, if $\mathcal{F}$ is finite, then*

$$\log f(z) \leq \log f_z(z) \leq \log f(z) + \log|\mathcal{F}| + \log\frac{1}{\delta},$$

*Further is the distribution family is finite, then*

$$\log f(z) \leq \log f_z(z) \leq \log f(z) + \log|\mathcal{F}| + \log\frac{1}{\delta}.$$

*Proof.* Let $K = \frac{1}{\delta} \sum_{z \in \mathcal{Z}} f_z(z)$.

$$\Pr(f(z) \leq f_z(z)/K) = \Pr\left(\frac{f_z(z)}{Kf(z)} \geq 1\right)$$
$$\overset{(a)}{\leq} \mathbb{E}\left[\frac{f_z(z)}{Kf(z)}\right]$$
$$= \frac{1}{K} \sum_{z \in \mathcal{Z}} f_z(z) = \delta,$$

where $(a)$ follows from Markov's inequality. The first part of the lemma follows by taking the logarithm and substituting the value of $K$. For the second part, observe that if $\mathcal{F}$ is finite then,

$$\sum_{z \in \mathcal{Z}} f_z(z) \leq \sum_{z \in \mathcal{Z}} \sum_{f \in \mathcal{F}} f(z) = \sum_{f \in \mathcal{F}} \sum_{z \in \mathcal{Z}} f(z) = \sum_{f \in \mathcal{F}} 1 = |\mathcal{F}|.$$

$\square$

We now prove a result of Shtarkov's sum, which will be useful in other scenarios.

**Lemma 3.** *Let $w = \pi(z)$ for some property $\pi$.*

$$\sum_w f_w(w) \leq \sum_z f_z(z).$$

*Proof.*

$$\sum_z f_z(z) \geq \sum_z p_{g(z)}(z) = \sum_w \sum_{z:g(z)=w} f_w(z) = \sum_w f_w(w).$$

$\square$

## C  GENERAL FIXED DESIGN RESULT

In this section, we will prove Theorem 2 which is an application of Theorem 1 to the fixed design regression setting with the square loss as the target metric. The theorem holds under some reasonable assumptions. We will fist prove some intermediate lemmas.

**Lemma 4.** *Let $f(\cdot; \boldsymbol{\theta}) = \prod_{i=1}^n p(\cdot|\boldsymbol{x}_i; \boldsymbol{\theta})$ in the fixed design setting. If $\|\boldsymbol{\theta} - \boldsymbol{\theta}'\|_2 \leq \delta$ then,*

$$\|f(\cdot; \boldsymbol{\theta}) - f(\cdot; \boldsymbol{\theta}')\|_1 \leq \sqrt{2nL\delta}.$$

*under the assumptions of Theorem 2.*

*Proof.* By Pinskers' inequality we have the following chain,

$$\|f(\cdot; \boldsymbol{\theta}) - f(\cdot; \boldsymbol{\theta}')\|_1 \leq \sqrt{2D_{\mathrm{KL}}(f(\cdot; \boldsymbol{\theta}); f(\cdot; \boldsymbol{\theta}'))}$$
$$\leq \sqrt{2 \sum_{i=1}^n D_{\mathrm{KL}}(p(\cdot|\boldsymbol{x}_i; \boldsymbol{\theta}); p(\cdot|\boldsymbol{x}_i; \boldsymbol{\theta}'))}$$
$$\leq \sqrt{2nL\delta}$$

$\square$

**Lemma 5.** *Assume the conditions of Theorem 2 holds. Let $\tilde{\boldsymbol{\theta}}_{\mathrm{mle}} := \mathrm{argmax}_{\boldsymbol{\theta} \in \mathcal{N}(\tilde{\epsilon}, \Theta)} \mathcal{L}(y^n, \boldsymbol{\theta})$, in the Poisson regression setting. Then with probability at least $1 - \delta$ we have,*

$$|\mathcal{E}(\hat{\boldsymbol{\theta}}_{\mathrm{mle}}) - \mathcal{E}(\tilde{\boldsymbol{\theta}}_{\mathrm{mle}})| \leq \frac{4w\lambda_{\max}(\Sigma)\tilde{\epsilon}}{n} \sqrt{\frac{\beta}{\alpha}}.$$

*Proof.* We assume that the event in (2) of Theorem 2 holds.

$$|\mathcal{L}(y^n; \hat{\boldsymbol{\theta}}_{\mathrm{mle}}) - \mathcal{L}(y^n; \boldsymbol{\theta}_c)| \leq \frac{\beta}{2}\tilde{\epsilon}^2,$$

where $\boldsymbol{\theta}_c$ is the closest point in $\mathcal{N}(\tilde{\epsilon}, \Theta)$ to $\hat{\boldsymbol{\theta}}_{\mathrm{mle}}$. Therefore, by definition

$$\mathcal{L}(y^n; \tilde{\boldsymbol{\theta}}_{\mathrm{mle}}) \leq \mathcal{L}(y^n; \hat{\boldsymbol{\theta}}_{\mathrm{mle}}) + (\beta/2)\tilde{\epsilon}^2.$$

By virtue of the strong-convexity we have,

$$\left\|\hat{\boldsymbol{\theta}}_{\mathrm{mle}} - \tilde{\boldsymbol{\theta}}_{\mathrm{mle}}\right\|_2^2 \leq \frac{\beta}{\alpha}\tilde{\epsilon}^2.$$

Now by triangle inequality we have,

$$|\sqrt{\mathcal{E}(\hat{\boldsymbol{\theta}}_{\mathrm{mle}})} - \sqrt{\mathcal{E}(\tilde{\boldsymbol{\theta}}_{\mathrm{mle}})}| = \left|\left\|\hat{\boldsymbol{\theta}}_{\mathrm{mle}} - \boldsymbol{\theta}^*\right\|_\Sigma - \left\|\tilde{\boldsymbol{\theta}}_{\mathrm{mle}} - \boldsymbol{\theta}^*\right\|_\Sigma\right|$$

$$\leq \left\|\hat{\boldsymbol{\theta}}_{\mathrm{mle}} - \tilde{\boldsymbol{\theta}}_{\mathrm{mle}}\right\|_\Sigma \leq \sqrt{\frac{\lambda_{\max}(\Sigma)\beta}{\alpha}}\tilde{\epsilon}$$

$\square$

Now we can use the fact $|a - b| \leq 2\max(\sqrt{a}, \sqrt{b})|\sqrt{a} - \sqrt{b}|$ and that $\Theta \subseteq \mathbb{B}_w^d$ to conclude,

$$|\mathcal{E}(\hat{\boldsymbol{\theta}}_{\mathrm{mle}}) - \mathcal{E}(\tilde{\boldsymbol{\theta}}_{\mathrm{mle}})| \leq 4w\lambda_{\max}(\Sigma)\tilde{\epsilon}\sqrt{\frac{\beta}{\alpha}}.$$

*Proof of Theorem 2.* Consider the net $\mathcal{N}(\phi, \theta)$. If $\phi = \delta^2/(2nL)$, then by Lemma 4 the net forms a $\delta$ cover on $\mathcal{F}$. Note that under the conditions of Theorem 2 $|\mathcal{N}(\phi, \theta)| \leq (3w/\phi)^d$.

Next note that in the application of Theorem 1, we should set

$$\epsilon = \frac{(c_1 + c_2\log(1/\delta))^\eta}{n}.$$

Thus if we set $\phi$ in place of $\tilde{\epsilon}$ in Lemma 5 we would need the following to apply Theorem 1,

$$4w\lambda_{\max}(\Sigma)\phi\sqrt{\frac{\beta}{\alpha}} = \frac{(c_1 + c_2\log(1/\delta))^\eta}{n}.$$

Thus, $\phi$ can be such that,

$$\log\frac{1}{\phi} \leq \max\left(\log(2nL), 0.5\log\frac{\beta}{\alpha} + \log(4w\lambda_{\max}(\Sigma)) + \log(n)\right).$$

From above we can apply Theorem 1 with the above $\epsilon$, $\delta_1 = \delta_2 = \delta$ and $T = (3w/\phi)^d$. Therefore, we get

$$\mathbb{P}\left(\mathcal{E}(\hat{\boldsymbol{\theta}}_{\mathrm{mle}}) \geq 3\frac{(c_1 + c_2\log(1/\delta))^\eta}{n}\right) \leq (T+5)\delta. \tag{5}$$

We can set $\delta = \delta/(T+5)$ to then conclude that w.p at least $1 - \delta$,

$$\mathcal{E}(\hat{\boldsymbol{\theta}}_{\mathrm{mle}}) = O\left(\frac{(c_1 + c_2 d(\log(w) + \log(1/\phi)))^\eta}{n}\right).$$

Substituting the bound on $\log\frac{1}{\phi}$ yields the result.

$\square$

# D  LEAST SQUARES FOR POISSON

We begin by restating the following general statement about OLS.

**Lemma 6** (see Theorem 2.2 in (Rigollet, 2015)). *Suppose $r = rank(\mathbf{X}^T\mathbf{X})$ and $\phi \in \mathbb{R}^{n \times r}$ be an orthonormal basis for the column-span of $\mathbf{X}$. Then we have the following result,*

$$\mathcal{E}(\hat{\boldsymbol{\theta}}_{\text{ls}}) \leq \frac{4}{n} \sup_{\boldsymbol{u} \in \mathbb{S}_1^{r-1}} (\boldsymbol{u}^T \tilde{\boldsymbol{\epsilon}})^2, \tag{6}$$

where $\tilde{\boldsymbol{\epsilon}} = \boldsymbol{\phi}^T \boldsymbol{\epsilon}$. Here, $\boldsymbol{\epsilon} = y^n - \mathbf{X}\boldsymbol{\theta}^*$ for the given response sample $y^n$.

Now are at a position to prove Lemma 1.

*Proof of Lemma 1.* We are interested in tail bounds on the RHS of (6). We will first analyze tail bounds for a fixed $\boldsymbol{u} \in \mathbb{S}_1^{r-1}$. We have the following chain,

$$\mathbb{E}[\exp\{s\langle \boldsymbol{u}, \tilde{\boldsymbol{\epsilon}}\rangle\}] = \mathbb{E}[\exp\{s\langle \boldsymbol{\phi}\boldsymbol{u}, \boldsymbol{\epsilon}\rangle\}] := \mathbb{E}[\exp\{s\langle \boldsymbol{v}, \boldsymbol{\epsilon}\rangle\}]$$

$$= \prod_{i=1}^{n} \mathbb{E}[\exp\{sv_i\epsilon_i\}] = \prod_{i=1}^{n} \mathbb{E}[\exp(sv_i(y_i - \mu_i))]$$

where $\mu_i = \langle \boldsymbol{\theta}^*, \boldsymbol{x}_i\rangle$. Now bounding each term separately we have,

$$\mathbb{E}[\exp(sv_i(y_i - \mu_i))] = \exp(-sv_i\mu_i)\exp(\mu_i(\exp(sv_i) - 1))$$

We have,

$$\prod_{i=1}^{n} \exp(\mu_i(\exp(sv_i) - 1)) = \exp\left(\sum_{i=1}^{n} \mu_i(\exp(sv_i) - 1)\right)$$

$$= \exp\left(\sum_{i=1}^{n}\left(\sum_{j=1}^{\infty} \frac{\mu_i}{j!}(sv_i)^j\right)\right) \leq \exp\left(\sum_{i=1}^{n}\left(\mu_i sv_i + \sum_{j=2}^{\infty} \mu_{max}\frac{1}{j!}(sv_i)^j\right)\right)$$

$$\leq \exp\left(\sum_{i} \mu_i sv_i + \mu_{max}(\exp(s) - s - 1)\right), \tag{7}$$

where we have used the fact that $||v||_p \leq ||v||_2 = 1$ for all $p \geq 2$. Thus we have,

$$\mathbb{E}[\exp\{s\langle \boldsymbol{u}, \tilde{\boldsymbol{\epsilon}}\rangle\}] \leq \exp(\mu_{max}(\exp(s) - s - 1)).$$

This means that the RV $\langle \boldsymbol{u}, \tilde{\boldsymbol{\epsilon}}\rangle$ is sub-exponential with parameter $\nu^2 = 2\mu_{max}$ and $\alpha = 0.56$. Thus if $t \leq 4\mu_{max}$ then we have that for a fixed $\boldsymbol{u}$,

$$\mathbb{P}(\langle \boldsymbol{u}, \tilde{\boldsymbol{\epsilon}}\rangle \geq t) \leq \exp\left(-\frac{t^2}{4\mu_{max}}\right). \tag{8}$$

Thus by a union bound over the $\epsilon$-net with $\epsilon = 2$, we have that wp $1 - \delta$,

$$\sup_{\boldsymbol{u}}\langle \boldsymbol{u}, \tilde{\boldsymbol{\epsilon}}\rangle \leq \sqrt{8\mu_{max}(\log(1/\delta) + r\log 6)}.$$

Thus we get the risk bound wp $1 - \delta$,

$$\mathcal{E}(\hat{\boldsymbol{\theta}}_{\text{ls}}) \leq \frac{32\mu_{max}}{n}(\log(1/\delta) + r\log 6). \tag{9}$$

For the sub-exponential region when $t \geq \mu_{max}$ we get the bound,

$$\mathcal{E}(\hat{\boldsymbol{\theta}}_{\text{ls}}) \leq \frac{16}{n}\max\left((\log(1/\delta) + r\log 6)^2, \mu_{max}^2\right).$$

We get the bound by setting $r = d$. $\qquad\square$

## E    COMPETITIVENESS FOR POISSON REGRESSION

**Lemma 7.** *Let $f(\cdot; \boldsymbol{\theta}) = \prod_{i=1}^{n} p(\cdot | \boldsymbol{x}_i; \boldsymbol{\theta})$ where $p$ is defined in Eq. (2). If $\|\boldsymbol{\theta} - \boldsymbol{\theta}'\|_2 \leq \delta$ then,*

$$\|f(\cdot; \boldsymbol{\theta}) - f(\cdot; \boldsymbol{\theta}')\|_1 \leq R\sqrt{\frac{2nw}{\gamma}}\delta.$$

*Proof.* By Pinskers' inequality we have the following chain,

$$\|f(\cdot; \boldsymbol{\theta}) - f(\cdot; \boldsymbol{\theta}')\|_1 \leq \sqrt{2D_{\mathrm{KL}}(f(\cdot; \boldsymbol{\theta}); f(\cdot; \boldsymbol{\theta}'))}$$

$$\leq \sqrt{2\sum_{i=1}^{n} D_{\mathrm{KL}}(p(\cdot|\boldsymbol{x}_i; \boldsymbol{\theta}); p(\cdot|\boldsymbol{x}_i; \boldsymbol{\theta}'))}$$

$$\leq \sqrt{2\sum_{i=1}^{n} (\langle \boldsymbol{\theta}, \boldsymbol{x}_i \rangle (\log(\langle \boldsymbol{\theta}, \boldsymbol{x}_i \rangle) - \log(\langle \boldsymbol{\theta}', \boldsymbol{x}_i \rangle)) - (\langle \boldsymbol{\theta}, \boldsymbol{x}_i \rangle - \langle \boldsymbol{\theta}', \boldsymbol{x}_i \rangle))}$$

Notice that the absolute value of the derivative of $a \log x - x$ wrt $x$, is upper bounded by $a/\gamma - 1$ if $x \geq \gamma > 0$ and also $a \geq \gamma$. Therefore, following from above we have,

$$\|f(\cdot; \boldsymbol{\theta}) - f(\cdot; \boldsymbol{\theta}')\|_1 \leq \sqrt{2\sum_{i=1}^{n} \left(\frac{\boldsymbol{\theta}^T \boldsymbol{x}_i}{\gamma} - 1\right) |\langle \boldsymbol{\theta}, \boldsymbol{x}_i \rangle - \langle \boldsymbol{\theta}', \boldsymbol{x}_i \rangle|}$$

$$\leq \sqrt{2\sum_{i=1}^{n} \left(\frac{\boldsymbol{\theta}^T \boldsymbol{x}_i}{\gamma} - 1\right) \|\boldsymbol{\theta} - \boldsymbol{\theta}'\|_2 R}$$

$$\leq \sqrt{\frac{2nwR^2}{\gamma} \|\boldsymbol{\theta} - \boldsymbol{\theta}'\|_2},$$

thus concluding the proof. □

Now we prove high probability bounds on the strong convexity and smoothness of the likelihood in the context of Poisson regression.

**Lemma 8.** *Let $\chi$ be the condition number of the matrix $\boldsymbol{M}$, where $\boldsymbol{u}_i = \boldsymbol{x}_i/\|\boldsymbol{x}_i\|_2$. Then with probability at least $1 - 2\delta$, we have*

$$\lambda_{\min}\left(\nabla_{\boldsymbol{\theta}}^2 \mathcal{L}(y^n; \boldsymbol{\theta})\right) \geq \frac{n}{2\|\boldsymbol{\theta}\|^2} \lambda_{\min}(\boldsymbol{M}), \tag{10}$$

*provided $n\lambda_{\min}(\boldsymbol{M}) \geq \frac{1}{4\gamma^2}(d\log(24\chi) + \log(1/\delta))$ and $\sqrt{\lambda_{\max}(\boldsymbol{M})(d\log(24\chi) + \log(1/\delta))} \leq \sqrt{n}\lambda_{\min}(\boldsymbol{M})/16$.*

*Proof.* We start with the expression of the Hessian of $\mathcal{L}(y^n; \boldsymbol{\theta})$,

$$\nabla_{\boldsymbol{\theta}}^2 \mathcal{L} = \sum_{i=1}^{n} \frac{y_i \boldsymbol{x}_i \boldsymbol{x}_i^T}{\langle \boldsymbol{\theta}, \boldsymbol{x}_i \rangle^2} \succcurlyeq \sum_{i=1}^{n} \frac{y_i \boldsymbol{u}_i \boldsymbol{u}_i^T}{\|\boldsymbol{\theta}\|^2} := L(\boldsymbol{\theta}),$$

where $\boldsymbol{u}_i = \boldsymbol{x}_i/\|\boldsymbol{x}_i\|_2$.

For a fixed unit vector $\boldsymbol{u} \in \mathbb{S}_1^{d-1}$ let us define,

$$L(\boldsymbol{u}, \boldsymbol{\theta}) := \sum_{i=1}^{n} \frac{y_i \boldsymbol{u}^T \boldsymbol{u}_i \boldsymbol{u}_i^T \boldsymbol{u}}{\|\boldsymbol{\theta}\|^2} := \sum_{i=1}^{n} y_i v_i,$$

where $v_i = \langle \boldsymbol{u}_i, \boldsymbol{u} \rangle^2 / \|\boldsymbol{\theta}\|^2$.

Following the definition of sub-exponential RV in (Rinaldo, 2019),

$$\sum_i y_i v_i \in SE\left(\nu^2 = \sum_i \nu_i^2, \alpha = 0.56 \max_i v_i\right), \tag{11}$$

where $\nu_i = 2\mu_i v_i^2$. This implies that wp atleast $1 - 2\delta/C$ we have,

$$|\sum_i y_i v_i - \sum_i \mu_i v_i| \leq 2\sqrt{\sum_i \mu_i v_i^2 \log\left(\frac{C}{\delta}\right)}$$

if,

$$\sqrt{\sum_i \mu_i v_i^2 \log\left(\frac{C}{\delta}\right)} \leq 2(\sum_i \mu_i v_i^2)/(\max_i v_i).$$

Note that the above condition is mild and is satisfied when $\lambda_{\min}\left(\sum_i \mu_i \boldsymbol{u}_i \boldsymbol{u}_i^T\right) \geq \log(C/\delta)/4\gamma^2$, as $\min_{\boldsymbol{u}} \sum_i \mu_i v_i \geq \lambda_{\min}\left(\sum_i \mu_i \boldsymbol{u}_i \boldsymbol{u}_i^T\right)$. Here, $C$ is a problem dependent constant that will be chosen later.

Now consider an $\epsilon$-net in $\ell_2$ norm over the surface unit sphere denoted by $\mathcal{N}(\epsilon, d)$. It is well known that $|\mathcal{N}(\epsilon, d)| \leq (3/\epsilon)^d$. Now any $\boldsymbol{z} \in \mathbb{S}_1^{d-1}$ can be written as $\boldsymbol{z} = \boldsymbol{x} + \boldsymbol{u}$ where $\boldsymbol{x} \in \mathcal{N}(\epsilon, d)$ and $\boldsymbol{u} \in \mathbb{B}_\epsilon^{d-1}$. Therefore, we have that

$$L(\boldsymbol{z}, \boldsymbol{\theta}) \geq L(\boldsymbol{x}, \boldsymbol{\theta}) - \lambda_{\max}(L(\boldsymbol{\theta}))\epsilon$$

A similar argument as above gives us,

$$\lambda_{\max}(L(\boldsymbol{\theta})) = \max_{\boldsymbol{u} \in \mathbb{B}_1^{d-1}} L(\boldsymbol{u}, \boldsymbol{\theta}) \leq \max_{\boldsymbol{x} \in \mathcal{N}(\epsilon, d)} L(\boldsymbol{x}, \boldsymbol{\theta}) + \frac{1}{2} \max_{\boldsymbol{u} \in \mathbb{B}_1^{d-1}} L(\boldsymbol{u}, \boldsymbol{\theta})$$

$$\implies \lambda_{\max}(L(\boldsymbol{\theta})) \leq 2 \max_{\boldsymbol{x} \in \mathcal{N}(\epsilon, d)} L(\boldsymbol{x}, \boldsymbol{\theta})$$

Thus by an union bound over the net we have wp $1 - 2\delta$ and other conditions,

$$\min_{\boldsymbol{z} \in \mathbb{S}_1^{d-1}} L(\boldsymbol{z}, \boldsymbol{\theta}) \geq \frac{1}{\|\theta\|_2^2} \lambda_{\min}\left(\sum_i \mu_i \boldsymbol{u}_i \boldsymbol{u}_i^T\right) - \frac{2\epsilon}{\|\theta\|_2^2} \lambda_{\max}\left(\sum_i \mu_i \boldsymbol{u}_i \boldsymbol{u}_i^T\right)$$

$$- \max_{\boldsymbol{u} \in \mathbb{S}_1^{d-1}} 2(1 + 2\epsilon) \sqrt{\sum_i \frac{\mu_i \langle \boldsymbol{u}, \boldsymbol{u}_i \rangle^2}{\|\boldsymbol{\theta}\|^4} \log\left(\frac{C}{\delta}\right)}.$$

if the conditions above hold with $C = (3/\epsilon)^d$. We can now set $\epsilon = 1/(8 * \chi)$ where $\chi$ is the condition number of the matrix $\sum_i \mu_i \boldsymbol{u}_i \boldsymbol{u}_i^T$. Now by virtue of that fact that $\sqrt{\lambda_{\max}(\sum \mu_i \boldsymbol{u}_i \boldsymbol{u}_i^T) \log(C/\delta)} \leq \lambda_{\min}\left(\sum_i \mu_i \boldsymbol{u}_i \boldsymbol{u}_i^T\right)/16$ we have the result. $\qquad \square$

Now we will prove a result on the smoothness of the negative log likelihood for the Poisson regression setting.

**Lemma 9.** *With probability at least $1 - 2\delta$, we have*

$$\lambda_{\max}\left(\nabla_{\boldsymbol{\theta}}^2 \mathcal{L}(y^n; \boldsymbol{\theta})\right) \leq \frac{2nR^2}{\gamma^2} \lambda_{\max}(\boldsymbol{M}), \tag{12}$$

*if $\lambda_{\min}(\boldsymbol{M}) \geq R^2(d \log 6 + \log(1/\delta))/4n\gamma^2$.*

*Proof.* We start by writing out the Hessian again but this time upper bounding it in semi-definite ordering,

$$\nabla_{\boldsymbol{\theta}}^2 \mathcal{L} = \sum_{i=1}^n \frac{y_i \boldsymbol{x}_i \boldsymbol{x}_i^T}{\langle \boldsymbol{\theta}, \boldsymbol{x}_i \rangle^2} \preccurlyeq \sum_{i=1}^n \frac{y_i \boldsymbol{x}_i \boldsymbol{x}_i^T}{\gamma^2} \preccurlyeq \sum_{i=1}^n \frac{y_i R^2 \boldsymbol{u}_i \boldsymbol{u}_i^T}{\gamma^2} := L(\boldsymbol{\theta})$$

For a fixed unit vector $\boldsymbol{u} \in \mathbb{S}_1^{d-1}$ let us define,

$$L(\boldsymbol{u}, \boldsymbol{\theta}) := \sum_{i=1}^n \frac{y_i R^2 \boldsymbol{u}^T \boldsymbol{u}_i \boldsymbol{u}_i^T \boldsymbol{u}}{\gamma^2} := \sum_{i=1}^n y_i v_i,$$

where $v_i = R^2 \langle \boldsymbol{u}_i, \boldsymbol{u} \rangle^2 / \gamma^2$. Proceeding as in Lemma 8 we conclude that,

$$\sum_i y_i v_i \in SE\left(\nu^2 = \sum_i \nu_i^2, \alpha = 0.56 \max_i v_i\right), \tag{13}$$

where $\nu_i = 2\mu_i v_i^2$. This implies that wp atleast $1 - 2\delta/C$ we have,

$$|\sum_i y_i v_i - \sum_i \mu_i v_i| \le 2\sqrt{\sum_i \mu_i v_i^2 \log\left(\frac{C}{\delta}\right)}$$

if,

$$\sqrt{\sum \mu_i v_i^2 \log\left(\frac{C}{\delta}\right)} \le 2(\sum \mu_i v_i^2)/(\max_i v_i).$$

Note that the above condition is mild and is satisfied when $\lambda_{\min}\left(\sum_i \mu_i \boldsymbol{u}_i \boldsymbol{u}_i^T\right) \ge R^2 \log(C/\delta)/4\gamma^2$, as $\min_{\boldsymbol{u}} \sum_i \mu_i v_i \ge \lambda_{\min}\left(\sum_i \mu_i \boldsymbol{u}_i \boldsymbol{u}_i^T\right)$. Here, $C$ is a problem dependent constant that will be chosen later.

Now consider an $1/2$-net in $\ell_2$ norm over the surface unit sphere denoted by $\mathcal{N}(0.5, d)$. It is well known that $|\mathcal{N}(\epsilon, d)| \le (3/\epsilon)^d$. Now any $\boldsymbol{z} \in \mathbb{S}_1^{d-1}$ can be written as $\boldsymbol{z} = \boldsymbol{x} + \boldsymbol{u}$ where $\boldsymbol{x} \in \mathcal{N}(0.5, d)$ and $\boldsymbol{u} \in \mathbb{B}_{0.5}^{d-1}$. Therefore, we have that,

$$\lambda_{\max}(L(\boldsymbol{\theta})) = \max_{\boldsymbol{u} \in \mathbb{B}_1^{d-1}} L(\boldsymbol{u}, \boldsymbol{\theta}) \le \max_{\boldsymbol{x} \in \mathcal{N}(\epsilon, d)} L(\boldsymbol{x}, \boldsymbol{\theta}) + \frac{1}{2} \max_{\boldsymbol{u} \in \mathbb{B}_1^{d-1}} L(\boldsymbol{u}, \boldsymbol{\theta})$$

$$\implies \lambda_{\max}(L(\boldsymbol{\theta})) \le 2 \max_{\boldsymbol{x} \in \mathcal{N}(\epsilon, d)} L(\boldsymbol{x}, \boldsymbol{\theta})$$

Thus we set $C = 6^d$ and obtain wp at least $1 - 2\delta$,

$$\lambda_{\max}(L(\boldsymbol{\theta})) \le 2\frac{nR^2}{\gamma^2} \lambda_{\max}(\boldsymbol{M}),$$

provided $4\sqrt{\lambda_{\max}(\boldsymbol{M}) \log(C/\delta)} \le \lambda_{\max}(\boldsymbol{M})/\sqrt{n}$. $\qquad\square$

*Proof of Corollary 1.* We show that the conditions of Theorem 2 hold.

*Creating Nets:* First we need to create an $\epsilon$-net over the parameter space $\Theta$. We start by creating an $\epsilon$-net over the sphere with radius $w$. Now suppose, $\epsilon < \gamma/2R$. Then we remove all centers $\boldsymbol{\theta}_c$ if $\exists i$ s.t $\langle \boldsymbol{\theta}_c, \boldsymbol{x}_i \rangle < \gamma/2$. This is a valid $\epsilon$-net over $\Theta$ as all net partitions that are removed do not have any points lying in $\Theta$. In subsequent section, we will always follow this strategy to create $\epsilon$-nets over subsets of $\Theta$.

*Strong Convexity and Smoothness:* From Lemma 8 and 9 we have that,

$$\frac{\beta}{\alpha} = \frac{w^2 R^2}{\gamma^2} \chi.$$

with probability $1 - O(\delta)$.

*KL Divergence:* The $L$ in Theorem 2 is bounded by $2wR^2/\gamma$ according to Lemma 7.

Combining the above into Theorem 2 and using the estimator in Section F we get our result.

$\qquad\square$

## F  MEDIAN OF MEANS ESTIMATOR FOR POISSON

In this section we will design a median of means estimator for the Poisson regression model based on the estimator proposed in the work of Lugosi & Mendelson (2019b). Recall that we have a fixed design matrix $\boldsymbol{X} \in \mathbb{R}^{n \times d}$ with rows $\boldsymbol{x}_1, \ldots, \boldsymbol{x}_n$, and for each $i \in [n]$, $y_i$ is drawn from a Poisson distribution with mean $\mu_i = \boldsymbol{\theta}^* \cdot \boldsymbol{x}_i$. We will further assume that the design matrix is chosen from an $(L, 4)$ hyper-contractive distribution. Mathematically this implies that for any unit vector $u$

$$\mathbb{E}[\left(\Sigma^{-\frac{1}{2}}\boldsymbol{x} \cdot u\right)^4] \leq L \cdot \mathbb{E}[\left(\Sigma^{-\frac{1}{2}}\boldsymbol{x} \cdot u\right)^2]^2.$$

For simplicity we will assume that $L = O(1)$. In the above definition note that $\mathbb{E}$ is the empirical expectation over teh fixed design. This is a benign assumption and for instance would be satisfied if the design matrix is drawn from a sub-Gaussian distribution. In the general case, the obtained bounds will scale with $L$. We have the following guarantee associated with Algorithm 1.

---

**Algorithm 1:** Median of Means Estimator

---

**Input:** Samples $S = \{(x_1, y_1), \ldots, (x_n, y_n)\}$, confidence parameter $\delta$.
**Step 1:** Compute $\Sigma = E[xx^\top]$. Form $S' = \{\boldsymbol{x}'_1 = y_1 \Sigma^{-\frac{1}{2}}\boldsymbol{x}_1, \ldots, \boldsymbol{x}'_n = y_n \Sigma^{-\frac{1}{2}}\boldsymbol{x}_n\}$.
**Step 2:** Randomly partition $S'$ into $k$ blocks of size $n/k$ each where $k = 20\lceil \log(\frac{1}{\delta})\rceil$.
**Step 3:** Feed in the k blocks to the median-of-means estimator of Lugosi & Mendelson (2019b) to get $\hat{\mathbf{v}}$.
**Step 4:** Return $\hat{\boldsymbol{\theta}} = \Sigma^{-\frac{1}{2}}\hat{\mathbf{v}}$.

---

**Theorem 4.** *There is an absolute constant $c > 0$ such that with probability at least $1 - \delta$, Algorithm 1 outputs $\hat{\boldsymbol{\theta}}$ such that*

$$\|\hat{\boldsymbol{\theta}} - \boldsymbol{\theta}^*\|_\Sigma^2 \leq c \cdot \|\boldsymbol{\theta}^*\|^2 \cdot \lambda_{max}(\Sigma)\left(\frac{d + \log(\frac{1}{\delta})}{n}\right). \tag{14}$$

*Proof.* The proof is exactly along the lines of the proof of Theorem 1 in the work of (Lugosi & Mendelson, 2019b) that we repeat here for the sake of completeness since the original theorem is not explicitly stated for a fixed design setting. To begin with notice that

$$\mathbb{E}[y\Sigma^{-\frac{1}{2}}\boldsymbol{x}] = \mathbb{E}[(\boldsymbol{\theta}^* \cdot \boldsymbol{x})\Sigma^{-\frac{1}{2}}\boldsymbol{x})] \tag{15}$$

$$= \mathbb{E}[(\boldsymbol{\theta}^* \cdot \boldsymbol{x})\Sigma^{-\frac{1}{2}}\boldsymbol{x}] \tag{16}$$

$$= \Sigma^{\frac{1}{2}}\boldsymbol{\theta}^*. \tag{17}$$

Hence if $\hat{\mathbf{v}}$ is the output of the median of the means estimator in Step 3 of Algorithm 1, then the least squares error of $\hat{\boldsymbol{\theta}}$ is exactly $\|\hat{\mathbf{v}} - \Sigma^{\frac{1}{2}}\boldsymbol{\theta}^*\|^2$. For convenience define $\boldsymbol{\mu}' = \Sigma^{\frac{1}{2}}\boldsymbol{\theta}^*$ and $\Sigma' = \mathbb{E}[\boldsymbol{x}'\boldsymbol{x}'^\top] - \Sigma^{\frac{1}{2}}\boldsymbol{\theta}^*\boldsymbol{\theta}^{*\top}\Sigma^{\frac{1}{2}}$. Exactly as in Lugosi & Mendelson (2019b) our goal is to show that $\boldsymbol{\mu}'$ beats any other vector $\boldsymbol{v}$ in the median of means tournament if $\boldsymbol{v}$ is far away from $\boldsymbol{\mu}'$. To quantify this define

$$r = \max\left(400\sqrt{\frac{Tr(\Sigma')}{n}}, 4\sqrt{10}\sqrt{\frac{\lambda_{\max}(\Sigma')}{n}}\right).$$

For a fixed vector $\boldsymbol{v}$ of length $r$, and block $B_j$, $\boldsymbol{\mu}'$ beats $v$ if

$$-\frac{2k}{n}\sum_{i \in B_j}(\boldsymbol{x}'_i - \boldsymbol{\mu}') \cdot \boldsymbol{v} + r > 0.$$

Let us denote by $\sigma_{i,j} \in \{0, 1\}$ a random variable representing whether data point $i$ is in block $j$ or not. By Chebychev's inequality we get that with probability at least $9/10$,

$$\left|\frac{k}{n}\sum_{i \in B_j}(\boldsymbol{x}'_i - \boldsymbol{\mu}') \cdot \boldsymbol{v}\right| \leq \frac{k}{n}\sqrt{10}\sqrt{\sum_i \mathbb{E}[\sigma_{i,j}^2((\boldsymbol{x}'_i - \boldsymbol{\mu}') \cdot \boldsymbol{v})^2]} \tag{18}$$

$$= \frac{k}{n}\sqrt{10}\sqrt{np\frac{1}{n}\sum_i \mathbb{E}[((\boldsymbol{x}'_i - \boldsymbol{\mu}') \cdot \boldsymbol{v})^2]}. \tag{19}$$

Here $p$ is the probability of a point belonging to block $k$. Noting that $np = \Theta(n/k)$ we get that with probability at least $9/10$,

$$\left|\frac{k}{n}\sum_{i \in B_j}(\boldsymbol{x}_i' - \boldsymbol{\mu}') \cdot \boldsymbol{v}\right| \leq \sqrt{10}r\sqrt{\frac{k\lambda_{\max}(\Sigma')}{n}}. \tag{20}$$

Applying binomial tail estimates we get that with probability at least $1 - e^{-k/180}$, $\boldsymbol{\mu}'$ beats $\boldsymbol{v}$ on at least $8/10$ of the blocks. By applying the covering argument verbatim as in the proof of Theorem 1 in Lugosi & Mendelson (2019b) we get that with probability at least $1 - \delta$, $\hat{\boldsymbol{v}}$ will satisfy

$$\|\hat{\boldsymbol{v}} - \Sigma^{\frac{1}{2}}\boldsymbol{\theta}^*\|^2 \leq c \cdot \lambda_{\max}(\Sigma')\left(\frac{d + \log(\frac{1}{\delta})}{n}\right).$$

Finally, it remains to bound the spectrum of $\Sigma'$. We have

$$\Sigma' = \mathbb{E}[y^2\Sigma^{-\frac{1}{2}}\boldsymbol{x}\boldsymbol{x}^\top\Sigma^{-\frac{1}{2}}] - \Sigma^{\frac{1}{2}}\boldsymbol{\theta}^*\boldsymbol{\theta}^{*\top}\Sigma^{\frac{1}{2}} \tag{21}$$

$$\preceq \mathbb{E}[((\boldsymbol{\theta}\cdot\boldsymbol{x})^2 + \boldsymbol{\theta}\cdot\boldsymbol{x})\Sigma^{-\frac{1}{2}}\boldsymbol{x}\boldsymbol{x}^\top\Sigma^{-\frac{1}{2}}] \tag{22}$$

$$\preceq T_1 + T_2, \tag{23}$$

where

$$T_1 = \mathbb{E}[(\boldsymbol{\theta}^* \cdot \boldsymbol{x})^2\Sigma^{-\frac{1}{2}}\boldsymbol{x}\boldsymbol{x}^\top\Sigma^{-\frac{1}{2}}], \tag{24}$$

$$T_2 = \mathbb{E}[(\boldsymbol{\theta}^* \cdot \boldsymbol{x})\Sigma^{-\frac{1}{2}}\boldsymbol{x}\boldsymbol{x}^\top\Sigma^{-\frac{1}{2}}]. \tag{25}$$

To bound $T_1, T_2$ we note that for any function $m(x)$ we have

$$\mathbb{E}[m(x)\Sigma^{-\frac{1}{2}}\boldsymbol{x}\boldsymbol{x}^\top\Sigma^{-\frac{1}{2}}] \preceq \sqrt{\mathbb{E}[m^2(x)]}I. \tag{26}$$

Using the above inequality and the fact that the design matrix is $(O(1), 4)$ hyper-contractive we get that

$$\sqrt{E[(\boldsymbol{\theta}^* \cdot \boldsymbol{x})^2]} = O(\|\boldsymbol{\theta}^*\|\sqrt{\lambda_{\max}(\Sigma)}) \tag{27}$$

$$\sqrt{E[(\boldsymbol{\theta}^* \cdot \boldsymbol{x})^4]} = O(\|\boldsymbol{\theta}^*\|^2\lambda_{\max}(\Sigma)). \tag{28}$$

Combining the above we get that

$$\Sigma' \preceq T_1 + T_2 \tag{29}$$

$$\preceq O(\lambda_{\max}(\Sigma))I. \tag{30}$$

$$\square$$

# G  1-D POISSON REGRESSION

When the covariates are one dimensional, a sharper analysis can actually be performed to show that the MLE dominates TMO in all regimes in the Poisson regression setting considered above.

**Lemma 10.** *There exists an absolute constant $c \geq 2$ such that for any $\delta \in [\frac{1}{n^c}, 1)$, it holds with probability at least $1 - \delta$ that,*

$$\mathcal{E}(\hat{\theta}_{\mathrm{ls}}) = \frac{1}{n}\sum_{i=1}^n(y_i - \hat{\theta}_{\mathrm{ls}})^2 \leq \frac{4 \cdot |\theta^*|}{n} \cdot \frac{\sum_{i=1}^n |x_i|^3}{\sum_{i=1}^n x_i^2}\log\left(\frac{1}{\delta}\right). \tag{31}$$

*The bound above is also tight i.e with constant probability it holds that*

$$\mathcal{E}(\hat{\theta}_{\mathrm{ls}}) = \frac{1}{n}\sum_{i=1}^n(y_i - \hat{\theta}_{\mathrm{ls}})^2 = \Omega\left(\frac{|\theta^*|}{n} \cdot \underbrace{\frac{\sum_{i=1}^n |x_i|^3}{\sum_{i=1}^n x_i^2}}_{:=\mathcal{B}(\hat{\theta}_{\mathrm{ls}})}\right). \tag{32}$$

We provide the proofs in later in the section. Having established the bound for least squares estimator, we next prove the following upper bound on the mean squared error achieved by the MLE.

**Theorem 5.** *There exists an absolute constant $c \geq 2$ such that for any $\delta \in [\frac{1}{n^c}, 1)$, it holds with probability at least $1 - \delta$ that,*

$$\mathcal{E}(\hat{\theta}_{\mathrm{mle}}) = \frac{1}{n}\sum_{i=1}^{n}(y_i - \hat{\theta}_{\mathrm{mle}})^2 \leq \frac{4 \cdot |\theta^*|}{n} \cdot \Big( \underbrace{\frac{\sum_{i=1}^{n} x_i^2}{\sum_{i=1}^{n} |x_i|}}_{:=\mathcal{B}(\hat{\theta}_{\mathrm{mle}})} \Big) \log\Big(\frac{2}{\delta}\Big). \tag{33}$$

It is easy to see that the covariate dependent term in the bound on the mean squared error achieved by $\hat{\theta}_{\mathrm{mle}}$ (defined in Eq. 32) is always better the corresponding term in the bound achieved by $\hat{\theta}_{\mathrm{ls}}$ (defined in Eq. 33). To see this notice that,

$$\frac{\mathcal{B}(\hat{\theta}_{\mathrm{ls}})}{\mathcal{B}(\hat{\theta}_{\mathrm{mle}})} = \frac{(\sum_{i=1}^{n} |x_i|^3)(\sum_{i=1}^{n} |x_i|)}{(\sum_{i=1}^{n} x_i^2)^2} \geq 1. \qquad \text{(from Cauchy-Schwarz inequality.)}$$

Furthermore, in many cases the bound achieved by $\hat{\theta}_{\mathrm{mle}}$ can be significantly better than the one achieved by $\hat{\theta}_{\mathrm{ls}}$. As an example consider a skewed data distribution where $\sqrt{n}$ of the $x_i$'s take a large value of $\sqrt{n}$, while the remaining data points take a value of $n^\epsilon$, where $\epsilon$ is a small constant. In this case we have that,

$$\frac{\mathcal{B}(\hat{\theta}_{\mathrm{ls}})}{\mathcal{B}(\hat{\theta}_{\mathrm{mle}})} = \frac{(\sum_{i=1}^{n} x_i^3)(\sum_{i=1}^{n} x_i)}{(\sum_{i=1}^{n} x_i^2)^2} = \Omega(n^\epsilon).$$

*Proof of Lemma 10.* Notice that

$$\mathcal{E}(\hat{\theta}_{\mathrm{ls}}) = \frac{1}{n}(\hat{\theta}_{\mathrm{ls}} - \theta^*)^2 (\sum_{i=1}^{n} x_i^2). \tag{34}$$

Hence, it is enough to bound the parameter distance, i.e., $(\hat{\theta}_{\mathrm{ls}} - \theta^*)^2$. In order to do that we first notice that $y_i x_i$ is a sub-exponential random variable with parameters $\nu_i^2, \alpha$ where where $\nu_i^2 = 2\mu_i x_i^2$, $\mu_i = \theta^* x_i$, and $\alpha_i = 0.56 x_i$ (Rinaldo, 2019). In other words,

$$y_i x_i \in SE(\nu_i^2, \alpha), \tag{35}$$

Thus from the bound on a sum of independent sub-exponential variables we have that,

$$\sum_i y_i x_i \in SE(\nu^2 = \sum_i \nu_i^2, \alpha = 0.56 \max_i x_i). \tag{36}$$

Thus we have that,

$$\mathbb{P}\Big( |\sum_i y_i x_i - \sum_i \mu_i x_i| \geq t \Big) \leq \begin{cases} 2\exp\Big(-\frac{t^2}{2\nu^2}\Big) & \text{if } t \leq \frac{\nu^2}{\alpha} \\ 2\exp\Big(-\frac{t}{2\alpha}\Big) & \text{otherwise} \end{cases}$$

This means, that w.p at least $1 - 2\delta$,

$$|\sum_i y_i x_i - \sum_i \mu_i x_i| \leq 2\sqrt{(\sum_i \mu_i x_i^2) \log\Big(\frac{1}{\delta}\Big)}$$

if

$$\sqrt{(\sum_i \mu_i x_i^2) \log\Big(\frac{1}{\delta}\Big)} \leq \frac{2\sum \mu_i x_i^2}{0.56 \max_i x_i}$$

The condition above is satisfied under our assumptions on $x_i$ and $\delta$, thereby leading to the bound

$$(\hat{\theta}_{\mathrm{ls}} - \theta^*) \leq 2\sqrt{w \log(1/\delta)} \frac{\sqrt{\sum x_i^3}}{\sum_i x_i^2}.$$

The bound on the MSE claimed in the lemma then follows.

Now we prove the lower bound. Again it is enough to show a lower bound on $|\hat{\theta}_{ls} - \theta^*|$. Notice that

$$\hat{\theta}_{ls} - \theta^* = \frac{\sum_{i=1}^n (y_i - \mu_i)x_i}{\sum_{i=1}^n x_i^2}. \tag{37}$$

Define the random variable $Z = \sum_{i=1}^n (y_i - \mu_i)x_i$. We will show anti-concentration for $Z$ by computing the hyper-contractivity of the random variable. Recall that a random varibale $Z$ is $\eta$-HC (hyper-contractive) if $\mathbb{E}[Z^4] \le \eta^4 \mathbb{E}[Z^2]^2$. Next we have

$$\mathbb{E}[Z^2] = \mathbb{E}[\sum_{i=1}^n (y_i - \mu_i)x_i]^2 = \sum_{i=1}^n \mu_i x_i^2. \tag{38}$$

$$\mathbb{E}[Z^4] = \mathbb{E}[\sum_{i=1}^n (y_i - \mu_i)x_i]^2$$

$$= \sum_{i=1}^n \mu_i(1 + 3\mu_i)x_i^4 + 2\sum_{i \ne j} \mu_i \mu_j x_i^2 x_j^2$$

$$:= \Delta. \tag{39}$$

Hence $Z$ is $\eta$-HC with $\eta^4 = \frac{\left(\sum_{i=1}^n \mu_i x_i^2\right)^2}{\Delta}$.

From anti-concentration of hyper-contractive random variables (O'Donnell, 2014) we have

$$\mathbb{P}(|Z| \ge \frac{1}{2}\sqrt{\mathbb{E}[Z^2]}) \ge \Omega(\eta^4). \tag{40}$$

Hence we get that

$$\mathbb{P}\left(|\sum_{i=1}^n (y_i - \mu_i)x_i| \ge \frac{1}{2}\sqrt{\sum_{i=1}^n \mu_i x_i^2}\right) \ge \Omega(\eta^4)$$

$$\ge \frac{\left(\sum_{i=1}^n \mu_i x_i^2\right)^2}{\Delta} \tag{41}$$

$$\tag{42}$$

Next notice that since $\mu_i \ge \gamma$ for all $i$ (Assumption 1), we have that $1 + 3\mu_i \le (3 + \frac{1}{\gamma})\mu_i$. This implies that $\Delta \le (3 + \frac{1}{\gamma})\left(\sum_{i=1}^n \mu_i x_i^2\right)^2$. Hence if $\gamma$ is a constant then with probability at least $\frac{1}{3 + \frac{1}{\gamma}} = \Omega(1)$ we have

$$\mathcal{E}(\hat{\theta}_{ls}) = \Omega\left(\frac{1}{n} \cdot \frac{(\sum_{i=1}^n x_i^3)}{\sum_{i=1}^n x_i^2}\right). \tag{43}$$

$\square$

*Proof of Theorem 5.* Recall that $\hat{\theta}_{mle}$ is defined as

$$\hat{\theta}_{mle} = \underset{\theta \in \Theta}{\operatorname{argmin}} \sum_{i=1}^n \theta x_i - y_i \log(\theta x_i). \tag{44}$$

Setting the gradient of the objective to zero, we get the following closed form expression for $\hat{\theta}_{mle}$.

$$\hat{\theta}_{mle} = \frac{\sum_{i=1}^n y_i}{\sum_{i=1}^n x_i}. \tag{45}$$

$\square$

Next, we note that $Z = \sum_{i=1}^{n} y_i$ is a poission random random variable with parameter $\mu = \sum_{i=1}^{n} \theta^* x_i$. From tail bounds for Poisson random variables (Klar, 2000) we have that for any $\epsilon > 0$,

$$\mathbb{P}[|Z - \mu| > \epsilon] \leq 2e^{-\frac{\epsilon^2}{\mu + \epsilon}}. \tag{46}$$

Taking $\epsilon = c\sqrt{\mu}\log(\frac{1}{\delta})$, we get that with probability at least $1 - \delta$,

$$|\sum_{i=1}^{n} y_i - \mu| \in \left(\frac{1}{2}, 2\right)\sqrt{\mu \log(\frac{1}{\delta})}, \tag{47}$$

provided that $\log(\frac{1}{\delta}) < \mu$ (that holds for our choice $\delta$, once $n$ is large enough). Hence we conclude that with probability at least $1 - \delta$, the mean squared error of $\hat{\theta}_{\mathrm{mle}}$ is bounded by

$$\begin{aligned}
|\hat{\theta}_{\mathrm{mle}} - \theta^*| &= \left|\frac{\sum_{i=1}^{n} \theta^* x_i}{\sum_{i=1}^{n} x_i} - \frac{\sum_{i=1}^{n} y_i}{\sum_{i=1}^{n} x_i}\right| \\
&= O\left(\frac{\sqrt{\mu \log(\frac{1}{\delta})}}{\sum_{i=1}^{n} x_i}\right) \\
&= O\left(\frac{\sqrt{|\theta^*| \log(\frac{1}{\delta})}}{\sqrt{\sum_{i=1}^{n} x_i}}\right).
\end{aligned} \tag{48}$$

The bound on the mean squared error follows from the above.

## H    COMPETITIVENESS FOR PARETO REGRESSION

We verify the conditions of Theorem 2 for the Pareto regression setting.

**Lemma 11.** *Let $f(\cdot; \boldsymbol{\theta}) = \prod_{i=1}^{n} p(\cdot|\boldsymbol{x}_i; \boldsymbol{\theta})$ where $p$ is defined in Eq. (4). If $\|\boldsymbol{\theta} - \boldsymbol{\theta}'\|_2 \leq \delta$ then,*

$$\|f(\cdot; \boldsymbol{\theta}) - f(\cdot; \boldsymbol{\theta}')\|_1 \leq \sqrt{\frac{2bn\delta R}{\gamma}}$$

*Proof.* By Pinskers' inequality we have the following chain,

$$\begin{aligned}
\|f(\cdot; \boldsymbol{\theta}) - f(\cdot; \boldsymbol{\theta}')\|_1 &\leq \sqrt{2D_{\mathrm{KL}}(f(\cdot; \boldsymbol{\theta}); f(\cdot; \boldsymbol{\theta}'))} \\
&= \sqrt{2\sum_{i=1}^{n} D_{\mathrm{KL}}(p(\cdot|\boldsymbol{x}_i; \boldsymbol{\theta}); p(\cdot|\boldsymbol{x}_i; \boldsymbol{\theta}'))} \\
&\leq \sqrt{\sum_{i=1}^{n} 2b|\log m_i - \log m_i'|} \\
&\leq \sqrt{\sum_{i=1}^{n} \frac{2b|\langle \boldsymbol{\theta}, \boldsymbol{x}_i \rangle - \langle \boldsymbol{\theta}', \boldsymbol{x}_i \rangle|}{\gamma}} \\
&\leq \sqrt{\frac{2bn\|\boldsymbol{\theta} - \boldsymbol{\theta}'\|_2 R}{\gamma}}.
\end{aligned}$$

The second inequality follows from the fact that $\log x$ is Lipschitz with parameter $L$ if $x > 1/L$. The last inequality follows from Cauchy-Schwarz and the norm bound on $\boldsymbol{x}_i$'s. $\square$

Now we prove the rest of the conditions.

**Lemma 12.** *We have the following smoothness bound,*

$$\lambda_{\max}(\nabla_{\boldsymbol{\theta}}^2 \mathcal{L}) \leq \frac{(b-1)}{\gamma^2}\lambda_{\max}(\Sigma).$$

*Proof.* We start by writing out the Hessian,

$$\nabla_{\boldsymbol{\theta}}^2 \mathcal{L} = \sum_{i=1}^n \frac{(b-1)}{\langle \boldsymbol{\theta}, \boldsymbol{x}_i \rangle^2} \boldsymbol{x}_i \boldsymbol{x}_i^T := L(\boldsymbol{\theta})$$

Using the fact that $\langle \boldsymbol{\theta}, \boldsymbol{x}_i \rangle \geq \gamma$ gives us the result. □

**Lemma 13.** *We have the following strong convexity bound,*

$$\lambda_{\min}(\nabla_{\boldsymbol{\theta}}^2 \mathcal{L}) \geq \frac{(b-1)}{w^2 R^2} \lambda_{\min}(\Sigma).$$

*Proof.* It follows from the expression of the Hessian in Lemma 12 and using the fact $\langle \boldsymbol{\theta}, \boldsymbol{x}_i \rangle \leq \|\boldsymbol{\theta}\|_2 \|\boldsymbol{x}_i\|_2$ □

*Proof of Corollary 2.* We show that the conditions of Theorem 2 hold.

*Creating Nets:* First we need to create an $\epsilon$-net over the parameter space $\Theta$. We start by creating an $\epsilon$-net over the sphere with radius $w$. Now suppose, $\epsilon < \gamma/2R$. Then we remove all centers $\boldsymbol{\theta}_c$ if $\exists i$ s.t $\langle \boldsymbol{\theta}_c, \boldsymbol{x}_i \rangle < \gamma/2$. This is a valid $\epsilon$-net over $\Theta$ as all net partitions that are removed do not have any points lying in $\Theta$. In subsequent section, we will always follow this strategy to create $\epsilon$-nets over subsets of $\Theta$.

*Strong Convexity and Smoothness:* From Lemma 13 and 12 we have that,

$$\frac{\beta}{\alpha} = \frac{w^2 R^2}{\gamma^2} \zeta.$$

*KL Divergence:* The $L$ in Theorem 2 is bounded by $2bR/\gamma$ according to Lemma 11.

Combining the above into Theorem 2 and using the estimator in (Hsu & Sabato, 2016) we get our result.

□

# I    MORE ON EXPERIMENTS

We provide more experimental details in this section.

## I.1    METRICS

The metrics and loss functions used are as follows:

**MSE:** The metric is

$$\frac{1}{n} \sum_{i=1}^n (\hat{y}_i - y_i)^2.$$

RMSE is just the square-root of this metric.

**MAE:** The metric is

$$\frac{1}{n} \sum_{i=1}^n |\hat{y}_i - y_i|.$$

**WAPE:** The metric is

$$\frac{\sum_{i=1}^n |\hat{y}_i - y_i|}{\sum_{i=1}^n |y_i|}.$$

**MAPE:** The metric is

$$\frac{1}{n} \sum_{i:y_i \neq 0} \left| 1 - \frac{\hat{y}_i}{y_i} \right|.$$

**Quantile Loss:** The reported metrics in Table 3 are the normalized quantile losses defined as,

$$\sum_{i=1}^{n} \frac{2\rho(y_i - \hat{y}_i)\mathbb{I}_{y_i \geq \hat{y}_i} + 2\rho(\hat{y}_i - y_i)\mathbb{I}_{y_i < \hat{y}_i}}{\sum_{i=1}^{n} |y_i|}. \tag{49}$$

During training the unnormalized version is used for quantile regression.

**Huber Loss:** The loss is given by,

$$L_\delta(y, \hat{y}) = \begin{cases} \frac{1}{2}(y - \hat{y})^2, & \text{if } |y - \hat{y}| \leq \delta \\ \delta|y - \hat{y}| - \frac{\delta^2}{2}, & \text{otherwise}. \end{cases}$$

## I.2   MORE DETAILS ABOUT THE MIXTURE DISTRIBUTION

We use a mixture distribution between zero, a *continuous extension* of negative binomial (NB) distribution and Pareto. The continuous extension of negative binomial is such that the p.d.f at $y$ is proportional to,

$$p(y) \propto \frac{\Gamma(n + k)}{\Gamma(k + 1)\Gamma(n)}(1 - p)^n p^k$$

given parameters $n$ and $p$. That is, the p.m.f of a regular discrete NB distribution is written in terms of Gamma functions and then we extend that to non-integral points up to proportionality, such that the measure sums to 1. This definition of NB is the standard implementation in Tensorflow (Abadi et al., 2016; Dillon et al., 2017), in order to support both discrete and continuous data . It has been used in many regression datasets before, especially in the field of genomics where the collected data is a discrete continuous mixture (Robinson & Smyth, 2008; Chen et al., 2016; McCarthy et al., 2012).

*Why not use a discrete-continuous mixture of zero, discrete NB and Pareto?*

A mixture of these three distributions is a discrete continuous mixture, whose CDF is well-defined. It is possible to define a likelihood in a standard manner such that it has a component proportional to the pdf of Pareto at all points and to this we add dirac delta functions proportional to magnitude of the negative-binomial pmf at non-negative integral points. We have an extra mass at zero to account for the zero component. The sum of the dirac masses along with the integral of the Pareto density sums to one. However, such a likelihood is not very useful in practice. For instance, if the data is mostly continuous but not heavy tailed, the log-likelihood would mostly have the Pareto component (because non-integral points have no contribution from the other components) which is not a desirable outcome, as a NB distribution can better model sub-Gaussian and sub-Exponential data in terms of moments.

## I.3   MAPPING OF OUTPUTS FOR ZNBP

For the `ZNBP` model we require an output dimension size of 6. The first three dimensions are mapped through a `softmax` layer (Goodfellow et al., 2016) to mixture weights. The fourth dimension is mapped to the 'n' in negative-binomial likelihood through the link function,

$$\phi(x) = \begin{cases} x + 1, & \text{if } x > 0 \\ 1/(1 - x) & \text{otherwise} \end{cases}.$$

The fifth dimension is mapped to 'p' of the negative-binomial though the sigmoid function. The last dimension is mapped to the scale parameter for the Pareto component using the $\phi(x)$ link function above.

## I.4   HARDWARE

We use the Tesla V100 architecture GPU for our experiments. We use Intel Xeon Silver 4216 16-Core, 32-Thread, 2.1 GHz (3.2 GHz Turbo) CPU and our post-hoc inference for MLE is parallelized over all the cores.

### I.5 MORE DETAILS ABOUT INFERENCE FOR MLE

We follow the approach of monte-carlo sampling. For each inference sample $\boldsymbol{x}'$, we generate 10k samples from the learnt distribution $p(\cdot|\boldsymbol{x}';\hat{\boldsymbol{\theta}}_{\mathrm{mle}})$. Then we compute the correct statistics. For MSE, RMSE the statistic is just the mean and for WAPE, MAE it is the median. For a quantile, it is the corresponding quantile from the empirical distribution. For any loss of the form,

$$\ell(y, \hat{y}) = \left| 1 - \left( \frac{y}{\hat{y}} \right)^{\beta} \right|$$

the optimal statistic is the median from the distribution proportional to $y^{\beta} p(y|\boldsymbol{x}';\hat{\boldsymbol{\theta}}_{\mathrm{mle}})$ (Gneiting, 2011). Note that the MAPE falls under the above with $\beta = -1$ and relative error corresponds to $\beta = 1$. The statistic can be computed by importance weighing the empirical samples.

### I.6 MODELS, HYPERPARAMETERS AND TUNING

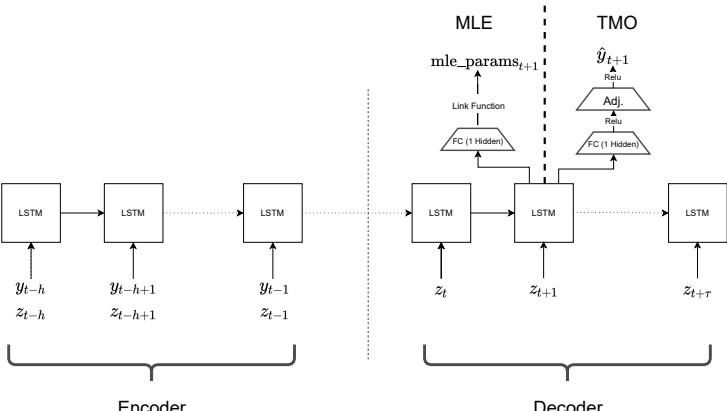

Figure 1: Time-Series Seq-2-Seq models. The MLE config is shown on the left and the TMO config is shown at the right. The main difference is the output dimension and the loss function. In order to keep the number of parameters the same, in the TMO model we add an extra layer of size 6 with Relu activation (shown as Adj. (adjustment))

For the time-series datasets, the model is a seq-2-seq model with one hidden layer LSTMs for both the encoder and the decoder. The hidden layer size is $h = 256$. The output layer of the LSTM is connected to the final output layer through one hidden layer also having $h$ neurons. Note that $h$ was tuned in the set [8, 16, 32, 64, 128, 256] and for all datasets and models 256 was chosen. In order to be keep the number of parameters exactly the same, in the TMO models we add an extra layer with ReLU with 6 neurons before the output.

We tuned the learning rate for Adam optimizer in the range [1e-5, 1e-1] in log-scale. The batch-size was also tuned in [64, 128, 256, 512] and the Huber-$\delta$ in [$2^i$ for i in range(-8, 8)]. The learning rate was eventually chosen as 2.77e-3 for both datasets, as it was close to the optimal values selected for all baseline models. The batch-size was chosen to be 512 and the Huber-$\delta$ was 32 and 64 for M5 and Favorita respectively.

For the regression datasets the model is a DNN with one hidden layer of size 32. For the Bicycle dataset the categorical features had there own embedding layer. The features [season, month, weekday, weathersit] had embedding layer sizes [2,4,4,2].

We tuned the learning rate for Adam optimizer in the range [1e-5, 1e-1] in log-scale. The batch-size was also tuned in [64, 128, 256, 512] and the Huber-$\delta$ in [$2^i$ for i in range(-8, 8)]. The learning rate was eventually chosen as 3e-3 for the gas turbine dataset based on the best perforamce of the baseline models and 1.98e-3 for the gas turbine dataset. The batch-size was chosen to be 512 and the Huber-$\delta$ was 128 and 32 for Bicyle Share and Gas turbine respectively.

For the ZNBP model we also tune the $\alpha$ parameter in the pareto component between [3, 4, 5]. The value of 3 was selected for all datasets, except for gas turbine where we used $\alpha = 5$.

MLE
TMO

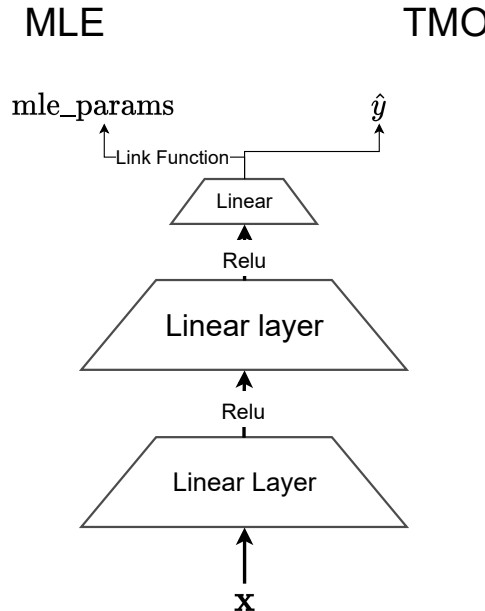

Figure 2: Fully connected network for regression models. The MLE config is shown on the left and the TMO config is shown at the right. The main difference is output dimension and the loss function.

| Dataset | $n$ | $d$ |
|---|---|---|
| M5 | 1879 | 256 |
| Favorita | 1653 | 256 |
| Bicycle | 584 | 32 |
| Gas Turbine | 22039 | 32 |

Table 5: Note that here $d$ refers to the dimension of the last layer of the architecture used in the respective datasets.

We used a batched version of GP-UCB (Srinivas et al., 2009) to tune the hyper-parameters. We used Tensorflow (Abadi et al., 2016) to train our models. In Table 5, we provide the number of samples and the dimension of the last layer in each of our datasets.

### I.7 DATASETS

For the Favorita and M5 dataset we used the product hierarchy over the item-level time series. Along with the item-level (leaf) time-series, we also add all the higher-level (parent) time-series from the product hierarchy (i.e. we add family and class level time-series for Favorita, and department and category level time series for M5). The time-series for a parent time-series is obtained as the mean of the time-series of its children. This is closer to a real forecasting setting in practice where one is interested in all levels of the hierarchy. The metrics reported are over all the time-series (both parents and leaves) treated equally. The history length for our predictions is set to 28.

For the M5 dataset the validation scores are computed using the predictions from time steps 1886 to 1899, and test scores on steps 1900 to 1913. For the Favorita dataset the validation scores are computed using the predictions from time steps 1660 to 1673, and test scores on steps 1674 to 1687.

The train test splits are as mentioned in the main paper. For the Gas turbine dataset we use the official train test split. For the Bicycle share data there is no official split, but we use a randomly chosen fixed 10% as the test set for all our experiments.

### I.8 ADDITIONAL EXPERIMENTS AND FIGURES

In order to show the dependency of $\lambda_{\max}(\Sigma)$ in the bound in Corollary 1 we perform a simulated experiment. We generate a dataset with $n = 5000$ and $d = 10$ such that each coordinate of $\mathbf{x}$ is distributed i.i.d from a uniform distribution between $[0, U]$. In out experiment, we vary $U$ in $\{1, 2, \cdots, 9\}$. Then $y$ is genearted from a Poisson distribution with rate $\langle \boldsymbol{\theta}^*, \mathbf{x} \rangle$ for a fixed $\boldsymbol{\theta}^*$. This varies $\lambda_{\max}(\Sigma)$ which is equal to $U^2/12$. We train and validate on 2500 samples with early stopping and plot the squared loss achieved on the test by the $\hat{\boldsymbol{\theta}}_{\text{mle}}$ based estimator in Figure 3 versus $\lambda_{\max}(\Sigma)$. We can clearly see a linear relationship which further validates our theoretical results.

In Figure 4, we plot the average training loss as a function of training iterations for the `MLE(ZNBP)` model. We can see that the loss converges to a minima.

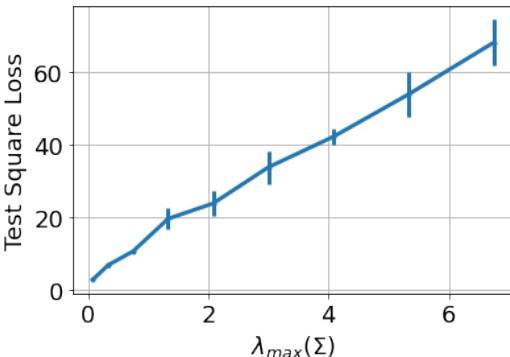

Figure 3: Test squared error versus $\lambda_{\max}(\Sigma)$. We can clearly see a linear relationship. Each point in the plot is averaged over 10 runs and we plot the standard error bars.

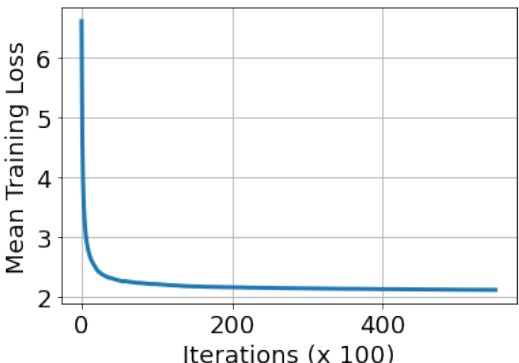

Figure 4: We plot the average training loss for the `MLE(ZNBP)` model as a function of training iterations. We can observe that the training curve converges.

## J EXTENDED DISCUSSION AND LIMITATIONS

We advocate for MLE with a suitably chosen likelihood family followed by post-hoc inference tuned to the target metric of interest, in favor of ERM directly with the target metric. On the theory side, we prove a general result that shows competitiveness of MLE with any estimator for a target metric under fairly general conditions. Application of the bound in the case of MSE for Poisson regression and Pareto regression is shown. We believe that our general result is of independent interest and can be used as a tool to prove competitiveness of MLE for a wide variety of problems. Such applications can be an interesting direction of future work.

On the empirical side we show that a well designed mixture likelihood like the one from Section 5 can adapt quite well to different datasets as the mixture weights are trained. As we have mentioned before,

the MLE log-likelihood loss in such cases can be non-convex which might lead to some limitations in terms of optimization. However, we observed that this is usually not a problem in practice and the solutions that can be reached by mini-batch SGD can be quite good in terms of performance.

In conclusion we would recommend the following protocol for a practitioner based on our theoretical and empirical observations:

If the overall problem is convex for TMO but introducing a MLE loss makes the problem non-convex, then the gains from the MLE approach might be neutralized by the added hardness of the non-convexity introduced. An example of such a situation is TMO for minimizing square loss on a linear function class, which is just least-squares linear regression, but introduction of a mixture likelihood like the one in Section 4 makes the problem non-convex. In this case it might be better to stick with TMO or at least proceed with caution with the MLE approach. Note that if the chosen MLE retains the convexity of the problem, for example Poisson MLE in Section 4.1, then we would still recommend going with the MLE approach.

However, in many practical scenarios when training using a deep network, the TMO approach is non-convex to begin with, even when the target metric itself is something simple and convex like the square loss. In such a case we would recommend the MLE approach with a likelihood class that can capture inductive biases about the dataset. This is because both TMO and MLE are non-convex and it is better to capitalize on the potential gains from the MLE approach.

Finally, note that the user can always choose between TMO and even between different likelihood classes through cross-validation in a practical setting. If the practitioner would like to forgo the decision making in choosing the likelihood class, we recommend using a versatile likelihood like the mixture likelihood in Sections 5.

We do not anticipate this work to have any negative social or ethical impact.

