# OpenReview forum: "On the benefits of maximum likelihood estimation for Regression and Forecasting"
_ICLR.cc/2022/Conference — ICLR 2022 Poster_

### Official Review · Reviewer_1pkV · 2021-10-29

**Correctness:** 3
**Technical Novelty And Significance:** 2
**Empirical Novelty And Significance:** 3
**Recommendation:** 5
**Confidence:** 2

**Main Review:**

Strengths:

- It is well known that the MLE is efficient under very mild assumptions, so although less intuitive compared with the TMO, it is important to promote its usage. One disadvantage of MLE is the likelihood is different to determine and it is often misspecified in practice. The paper discusses this point and recommend using a mixture of distributions.
- The finite sample error bound is obtained in a more general context than Acharya et al. (2017).
- Specific examples (Poisson regression and Pareto regression) are provided to illustrate the theoretical results.

Weaknesses:

- The notations and presentation are difficult to follow. Some key notations do not seem to be properly defined. Here are some examples. What is the meaning of $\ell_{1}$ norm for probability distributions? $f$ stands for the probability density function or cumulative distribution function or the underlying probability measure? What is the meaning of $|\tilde{F}|$? From Definition 1, it seems to be the size of the set, but this does not seem to be the case when I check the proof. Due to these issues, I was not able to fully understand and appropriately evaluate the significance of the theoretical results, and I have reflect this in my confidence level.
- The assumptions in Theorem 1 seems to already require that the MLE has a tight bound. I am not clear how to interpret the main result.
- I am not sure if it is true that the proof for discrete distributions can be easily extended to continuous distributions. This is usually not the case in existing MLE theory partly due to fact that the support for continuous distributions are not countable.
- Why both $\ell_{1}$ norm and KL-divergence are required.
- For the two distributions in section 4, it is well known that the MLE is more efficient than the least squares.
- Minor issues:
  - "w.p" should be "w.p." in multiple places
  - "s.t" should be "s.t."
  - The result in (Bai & Yin, 2008) for random covariance matrices is for Gaussian distributed covariates only.
  - It might be better to give the definition of MAPE, WAPE and RMSE in the main paper than in the appendix so that readers can follow the main paper without going back and forth.


**Summary Of The Paper:**

This paper promotes the maximum likelihood estimation over target metric optimization. The main theoretical results is a finite sample error bound. Specific examples and numerical results are provide to further illustrate the investigation.

**Summary Of The Review:**

This paper in about an interesting and important idea, but the notations and presentation need some improvement.

---

> ### Author Response · Authors · 2021-11-14
> **Response to the Review**
>
> 1. We apologize for not clarifying all the notations. $\ell_1$ norm is twice the total variational distance between the probability measures. We have added this in the paper. $f$ stands for the probability density function. |.| denotes the size of the set, so your interpretation is correct. Note that the cover $\tilde{\mathcal{F}}$ is finite as defined in Definition 1, even when $\mathcal{F}$ is not finite. In the proof of Theorem 1 the size is actually used in the last inequality and thereafter where $\sum_{\tilde{f} \in \tilde{F}} \delta$ term gives rise to the $|\tilde{F}| \delta$ term in the final result.
>
> 2. We would like to clarify that the theorem states that if we can find any estimator $\hat{\pi}(z)$ (not the MLE) with a tight bound, then the MLE would be competitive with it under our assumptions. So this part does not require a tight bound on the MLE, but instead on any other estimator of the problem. For instance, in our application in the Poisson setting, we derive a _complicated_ estimator with a tight bound (see Eq 3) and there after using Theorem 1 we are able to prove that MLE will be competitive with this estimator and therefore also has a good bound. The significance of Theorem 1 in words is that under reasonable conditions MLE is as good as any other possibly complicated estimator for any target metric, while having the advantage of not needing prior knowledge about the target metric
>
> 3. We have written the proof for the continuous case in Appendix A. All the steps go through. The only point of difference is that we need to invoke Fubini’s theorem to interchange the sum and the integral in step (d) of the proof of Theorem 1.
>
> 4. For Theorem 2 we only have KL divergence as a requirement as it is easy to bound in most cases and decomposes for product distributions, thus making the Theorem more easy to apply. In the proof we use Pinskers to bound $\ell_1$ by the KL divergence and thus are able to apply Theorem 1.
>
> 5. While it is indeed known that least squares performs poorly for the heavy tailed setting, we are not aware of finite sample results where it is shown that the MLE based estimator is better than least squares in terms of square loss for these two settings. Note that MLE has the advantage of using the correct distribution but least squares has the advantage of being guided by the target metric. So it is not immediately clear which one would be better. Therefore we believe there is value in proving these results, and they show the applicability of our main Theorems 1,2. Note that our results are finite samples and are not implied by asymptotic parameter recovery results on the MLE.
>
> 6. Minor issues: We thank the reviewer for these suggestions. We have incorporated them in the paper, however due to space limitations we were not able to move the definition of MAPE, WAPE etc to the main paper.

---

> > ### Author Response · Authors · 2021-11-29
> > **Response to Reviewer 1pkV**
> >
> > We thank you for your insightful comments and hope that we have addressed your concerns above. Please let us know if you have further questions, so that we have a chance to reply to them before the discussion phase ends tonight.

---

> > > ### Comment · Reviewer_1pkV · 2021-11-29
> > > **No further questions**
> > >
> > > Thank you for your detailed responses to my comments and questions. I do not have additional questions.

---

### Official Review · Reviewer_Fcze · 2021-11-05

**Correctness:** 3
**Technical Novelty And Significance:** 3
**Empirical Novelty And Significance:** 4
**Recommendation:** 5
**Confidence:** 3

**Details Of Ethics Concerns:**

I do not find any ethical issues with this paper.

**Main Review:**

**Strengths:**

(a) The paper tackles an interesting and challenging topic to compare two well-known inferential methods for regression, namely, the maximum likelihood estimation (MLE) and the estimation based on loss functions (TMO).

(b) A new theorem, Theorem 2, is given to compare the performance of the MLE with that of an estimator in terms of a quantity which measures the correctness of estimators of regression coefficients.

(c) Applying Theorem 2, detailed discussion is given to compare MLE and TMO for Poisson regression and Pareto regression.

**Weaknesses:**

(d) I wonder the mathematical correctness of the discussion in Section 5 related to the three component mixture distribution which consists of the point distribution at 0, the negative binomial distribution and Pareto distribution. The point distribution and negative binomial distribution are discrete distributions, while Pareto distribution is a continuous distribution. Therefore the likelihood function of the mixture of these distributions should not be evaluated based on the function $\sum_{j=1}^3 w_j p_j(y| \boldsymbol{x} ; \boldsymbol{\theta}_j)$, where $p_1$ and $p_2$ are the probability mass functions of the point distribution and the negative binomial distribution, respectively, and $p_3$ is the probability density function of the Pareto distribution. I am worried from the discussion in Section 5 that this approach is used for the maximum likelihood estimation of the mixture distribution. There should be more explanation about the maximum likelihood estimation of this mixture model.

(e) If my worry in Comment (e) is correct, part of the results of the experiments in Section 6 regarding the three component mixture should be reconsidered.

(f) In the paper, MLE and TMO are compared through the quantity given in Equation (1). However I am not sure how meaningful this comparison will be because MLE and TMO are generally the minimizers of loss functions which are not based on the square loss risk as in (1). Since the two estimators are obtained through different loss functions, I think it is difficult to compare the goodness of the estimators through the single quantity such as the quantity (1). Or is there any special meaning to use the quantity (1) to evaluate the MLE and TMO?

**Other Comments:**

(g) I am confused about the MLE for regression models used in the paper. In the ordinary MLE for regression, the parameters of a probability distribution and regression coefficients are estimated simultaneously via the maximization of the likelihood function. However it seems from the paragraph titled "MLE and post-hoc inference" on page 4 that the parameters of the distribution $\boldsymbol{\theta}$ and the link function $\tilde{h}$ are estimated in different steps. Is this approach used for Theorem 2 and other results of the paper? This should be explained in the paper because the approach discussed there is different from the common approach for MLE.

(h) p.2, Section 1, Competitiveness of MLE, l.2 up: MLE can competitive  ===> MLE can be competitive

(i) p.3, Section 3, Notation, ll.1-2 up: the sphere centered at ... ===> the ball centered at ... (I think this expression is less confusing.)

(j) p.3, Definition 1, l.3: The definition of $| \cdot |$ in $|\tilde{\cal F}| \leq T$ should be given.

(k) p.3, Theorem 1, l.2 up: Is "the MLE based estimator" exactly the same as the MLE or something different?

(l) p.6, Equation (4):  As a comment related to Comment (g), how is the parameter $b$ in the Pareto regression model (4) estimated?

(m) p.8, Section 6, Common Experimental Protocol, 2nd paragraph, l.2 up: that employs the mixture ===> that employ the mixture

(n) p.28, l.7 up: be keep the number ===> keep the number

**Summary Of The Paper:**

This paper compares two inferential methods for regression models, the maximum likelihood estimation and the estimation based on loss functions. For that purpose, a quantity is proposed to measure the correctness of an estimator of a regression model. Then it is shown that, under certain conditions on the quantity of an estimator, the proposed quantity of the maximum likelihood estimator can be evaluated. This result is applied to two regression models based on the Poisson distribution and Pareto distribution. Choices of the probability distributions and target metrics are discussed for the maximum likelihood estimation. Experiments are given to compare the performance of the maximum likelihood estimator and the estimators based on some loss functions.

**Summary Of The Review:**

The paper tackles a challenging topic to compare two well-known inferential methods for regression. Some new theoretical results are presented on this topic. However I have concerns about the mathematical correctness of the paper (see Comments (d) and (e)) and the motivation for the study (see Comment (f)).

---

> ### Author Response · Authors · 2021-11-14
> **Response to the Review (1/2)**
>
> (d) We thank the reviewer for pointing out the lack of clarity in Section 5. We have updated the paper with details about the log-likelihood function that we use.
>
> We wanted to design a likelihood that can model sparse data, different kinds of sub-gaussian and sub-exponential tails as well as heavier tails, and be generally applicable to real-valued datasets. Therefore we use a mixture distribution between zero, the __continuous extension__ of negative binomial distribution and Pareto. The continuous extension of negative binomial is such that the p.d.f at $y$ is proportional to,
> \begin{align*}
> p(y) \propto \frac{\Gamma(n + k)}{\Gamma(k + 1) \Gamma(n)} (1 - p)^{n}p^{k}
> \end{align*}
> given parameters $n$ and $p$. That is, the p.m.f of a regular discrete NB distribution is written in terms of Gamma functions and extend that to non-integral points up to proportionality, such that the measure sums to 1.  This definition of NB  is the standard implementation in Tensorflow Probability [1], in order to support both discrete and continuous data . It has been used in many regression datasets before, especially in the field of genomics where the collected data is a discrete continuous mixture [2, 3, 4].
>
> (e) Under the definition of our mixture above, the log-likelihood is a well defined loss function in terms of the learnable parameters and therefore our experiments in Section 6 are valid. The above mixture likelihood is used in practice because we found that it yields a loss function that works well for a variety of datasets as shown in Section 6.
>
> Note that our theoretical results only apply our general Theorem 1 to the Poisson regression and Pareto regression settings in Section 4. Extending these results to negative binomial and more generally to the specific mixture used in the experiments is an interesting direction of future work and is beyond the scope of this paper.
>
> (f) Our goal is to compare MLE and TMO methods in terms of a target metric on the test data. This target metric is the square loss (Eq. 1) for both the theoretical regression settings considered in Section 4. Therefore even though only TMO is trained using the squared loss, and MLE is trained using a different loss function  in our theory they are always compared on the basis of their expected performance in terms of _the same_ test metric (square loss) on the population level.  Therefore  this is a fair comparison (if at all, the TMO has a perceived advantage since it is trained using the same empirical loss as the target metric). We clearly define both the methods and the means of comparison in Section 3 (Page 4).
>
> In our experiments, we test both the methods on three different target metrics, for each dataset. The target metrics considered are MAPE, WAPE and RMSE.  MLE minimizes the negative log-likelihood, and predicts the statistic from the MLE distribution that minimizes each target metric (eg. for RMSE it's the mean and for WAPE it's the median). TMO directly minimizes the target metric on the training and predicts the output on the test samples. We showcase that for most of the target metrics the MLE method is superior, even though it only needs to train one model per dataset, while the TMO method essentially trains one model per target metric for each dataset. Note that as before the MLE and TMO models are always compared based on the same target metric on the test set and therefore the comparison is fair.
>
> (g, l) In the “MLE and post-hoc inference” paragraph in page 4, $\tilde{h}$ has no relation with the link function. The parametric form of the family $p(y; \mathcal{f}; \pmb{\theta})$ captures the link function as well as the learnable parameters. $\tilde{h}(\mathbf{x}’)$ denotes the prediction given a new sample $\mathbf{x}’$ which is basically the statistic from the learnt MLE distribution of $y$ given $\mathbf{x}’$ that minimizes the target metric. For example if $\ell$ is the square loss, this would be $\mathbb{E}[y | \mathbf{x}; \pmb{\theta}_{mle}]$.
>
> As discussed in the “common experimental protocol” section in page 8, for the MLE method the output layer is mapped to the parameters for the likelihood through fixed link functions (not learnt). The link functions are defined in Appendix I.2. To give an example, suppose we are training with log-likelihood of negative binomial then there are two parameters $(n(x), p(x))$ that define the distribution of $y| \mathbf{x}$. Then the output layer of the network is mapped by one linear layer followed by the link function $\phi(x)$ in section I.2 to $n$. It is also mapped by another linear layer followed by sigmoid function to $p$. All the weights are trained jointly. In the Pareto regression problem, for our analysis the task is to estimate the coefficients that appear in the scale parameter $m_i$. We assume that for this setting the true shape parameter $b$ is known.

---

> > ### Author Response · Authors · 2021-11-14
> > **Response to the Review (2/2)**
> >
> > In the summary the reviewer has raised concerns about the mathematical correctness of the paper and has also marked the option “Several of the paper’s claims are incorrect or not well-supported.” However, the issue raised by the reviewer is a definition issue in our empirical results section in Section 5 and 6. We have clarified this issue above, and in the updated paper, to explain why our experiments are indeed valid and this had no bearing on the correctness of our theoretical results in Section 3 and 4 to begin with. We hope we have clarified this confusion and we would be grateful if the reviewer re-evaluates the technical aspects of the paper.
> >
> > Other minor comments: We are grateful to the reviewer for pointing these out. We have corrected the typos in the updated version.
> >
> > [1] https://www.tensorflow.org/probability/api_docs/python/tfp/distributions/NegativeBinomial
> >
> > [2] Mark D Robinson and Gordon K Smyth. Small-sample estimation of negative binomial dispersion, with applications to sage data.Biostatistics, 9(2):321–332, 2008.
> >
> > [3] Yunshun Chen, Aaron TL Lun, and Gordon K Smyth. From reads to genes to pathways: differential expression analysis of rna-seq experiments using rsu bread and the edger quasi-likelihood pipeline. F1000Research, 5, 2016.
> >
> > [4] Davis  J  McCarthy,  Yunshun  Chen,  and  Gordon  K  Smyth.   Differential  expression  analysis  of multifactor rna-seq experiments with respect to biological variation.Nucleic acids research, 40(10):4288–4297, 2012

---

> > > ### Comment · Reviewer_Fcze · 2021-11-25
> > > **Re: Response to the Review**
> > >
> > > I would like to thank the authors for careful responses to my comments.
> > >
> > > I have a better understanding about the results of the paper after reading the responses. Regarding my comment (d), it is good to know that the authors actually consider a continuous version of the negative binomial distribution instead of the (original) discrete one. In addition, after reading the response to my comment (g), I understand that the parameter space $\Theta$ in the "MLE and post-hoc inference" section on page 4 refers to the parameter space of the regression coefficients.
> > >
> > > Considering these clarifications, I upgraded my score for recommendation and correctness. However I have to agree with another reviewer that the notations and presentation of the paper are difficult to follow. The current presentation of the paper easily leads to misunderstanding about the results. I think that the improvement in the presentation of the paper will improve the value of this paper significantly.

---

### Official Review · Reviewer_BdKr · 2021-11-07

**Correctness:** 3
**Technical Novelty And Significance:** 3
**Empirical Novelty And Significance:** 3
**Recommendation:** 8
**Confidence:** 3

**Main Review:**

The theoretical novelty of the paper lies in the fact that it is the first work that determines the performance of a MLE in terms of the performance of existing estimators using finite samples, as opposed to asymptomatic convergence to the true value. Subsequently, it derives tighter bounds, than existing results, for mle poisson and pareto regression.

(1) The results seem to hold in low-dimensional cases where (n -- samples> d- covariates). The practical utility of the paper would be stronger if it could be applied in the high-dimensional regime ( for example in corollary 2, the d^2 factor seems to be warning in this scenario).

(2) Similarly, it would be interesting if the authors could compare also with ridge/lasso estimators in section (6).

(3) The bound of corollary (2) is the tightest that can be achieved (one cannot find a different \theta_est satisfying the conditions of theorem (2) that could yield better excess square loss risk for the MLE?)

(4) Regarding the bound of lemma 1, and given that I am not an expert in the area, I cannot grasp its tightness (how large is \lambda_max (\Sigma) ) in practical settings. I think it would help if the authors could show experimentally ( for experiments that do not make use of the mixture of distributions suggested- but only the poisson and/or pareto distribution separately ) and demonstrate empirically the tightness of the bounds.

(5) Moreover, I think it would be useful if the authors could summarize in a Table n and d for the datasets considered.

(6) Given that the resulting loss  that stems from the proposed mixture is con-convex,  it would be useful if the authors could demonstrate convergence empirically by providing some learning curves in the appendix.

(7) Finally, it is not clear to me why theorem 2 can be applied on the mixture (even when it holds separately for each component).

**Summary Of The Paper:**

The paper provides theoretical results that favor MLE estimators, in terms of the excess square loss risk,  compared to empirical risk estimators under mild assumption. In particular, the paper devises an estimator for Poisson regression and employs an existing estimator for heavy-tail Pareto regression, and subsequently derives upper bounds on the excess square loss risk that are tighter compared to the least squares estimator. Finally, a mixture of distributions (with each component coming from a different distribution class) is proposed. The MLE for the proposed mixture outperforms empirical risk estimators under different objective losses.

**Summary Of The Review:**

The paper makes theoretical contributions on the performance of mle compared to empirical risk losses.
However, the applicability of the derived results in real-world regimes could be improved.

---

> ### Author Response · Authors · 2021-11-17
> **Response to the Review**
>
> We thank the reviewer for the comments and suggestions.
>
> 1. We thank the reviewer for this suggestion. There is a body of work [1, 2, 3, 4] that shows sub-optimality of MLE when $d$ grows with $n$. So we might need to make further structural assumptions to show such a result. It is an interesting future direction but beyond the scope of this paper.
>
> 2. In Section 6 the empirical results are for deep learning models where only the loss function is modified in the last layer i.e the only difference between TMO and MLE is the loss function. We are assuming that the reviewer means that we should tune the l1/l2 penalty on the last layer for TMO. We added an elastic net penalty on the last layer weights and tuned the l1 and l2 weights in [1e-5, 1e2] in log-scale. The results from the best model on the Favorita dataset are as follows:
> | Model | MAPE | WAPE | RMSE|
> |----------|----------|----------|----------|
> | TMO_Reg(MSE) | 0.6058+/-0.0067 | 0.2876+/-0.0016 |177.8805+/-0.7724 |
> | TMO_Reg(MAE) | 0.3949+/-0.0014 | 0.2244+/-0.0006 | 162.5849+/-0.5583 |
> | TMO_Reg(MAPE) | 0.3201+/-0.0009 | 0.2553+/-0.0009 | 199.251+/-0.7263 |
>
> We can see that the numbers are still worse than the MLE based model, and in fact there is no perceived benefit from l1/l2 regularization in this case.
>
> 3. This is a great point. There could indeed potentially be a different estimator that could be used to strengthen Corollary 2, but our motivation was to use Theorem 2 along with an estimator to show that MLE is better than TMO, which is already achieved by Corollary 2. This actually shows the strength of our competitive bounds, since if one finds a better estimator satisfying our conditions, it would immediately yield a better bound for the MLE estimator.
>
> 4. $\lambda_{max}(\Sigma)$ is a property of the averaged design matrix. For instance if the design matrix was random Gaussian, then the maximum eigen-value would be O(1) w.h.p [5]. In the worst case, $\lambda_{max}(\Sigma)$ is bounded by average $||x_i||^2$ which is reasonably small for most real datasets. We performed a simulation experiment to plot the test squared error vs $\lambda_{max}(\Sigma)$ in a Poisson regression setting. The results have been added to Appendix I.8 and they show a linear relationship, thus further validating our theoretical bounds.
>
> 5. We have added that as Table 5 in Appendix I. Please note that in our experiments we use deep networks, so we only report the dimensionality of the last linear layer.
>
> 6. Thanks for the suggestion. We have added that in the appendix.
>
> 7. Theorem 2 would not be directly applicable to the mixture since the log likelihood is non-convex for mixture distributions. The mixture distribution is used in our experiments as we have noticed that it  works well in practice for a large variety of datasets (please see the ablation study in Table 4). It would be an interesting question to prove a version of Theorem 2 for a mixture of Gaussians in a local region around the optima, but that is beyond the scope of this work.
>
> [1, 2, 3, 4] (Sur & Candès, 2019; Bean et al., 2013; Donoho & Montanari, 2016; El Karoui, 2018) as cited in the paper.
>
> [5] https://arxiv.org/abs/cond-mat/0701371

---

### Author Response · Authors · 2021-11-24
**Thank you for the comments and suggestions.**

We thank the reviewers for their insightful comments and suggestions. We hope that we have been able to address the concerns. Please let us know in case there are further questions.

---

### Decision · Program_Chairs · 2022-01-20

**Decision:**

Accept (Poster)

**Comment:**

This paper has been independently assessed by three expert reviewers. The results place it at the borderline of acceptance decision: while one of the reviewers gave it a straight accept evaluation, two others assessed it as marginally rejectable, even after discussion with the authors. All of the reviewers agreed that the theoretical results provided should help promote the use of MLE estimators over perhaps more prevalently used in current practice TMO, and that is the main contribution of this work. The reviewers were concerned with the clarity of the presentation and with a confusing notation used. Some of these issues have been addressed in the authors' responses. All things considered, I conclude that this work can be of some interest to the ICLR audience, and as such it can be assessed as marginally acceptable for this conference: "accept if needed". I will recommend it as such for consideration by the Senior Area Chair and the Program Committee.